# An ultrasensitive and broadband transparent ultrasound transducer for ultrasound and photoacoustic imaging in-vivo

Seonghee Cho [1,2,7], Minsu Kim [2,3,7], Joongho Ahn[2,3], Yeonggeun Kim [2,3], Junha Lim[4], Jeongwoo Park[2,3], Hyung Ham Kim[1,2,3], Won Jong Kim [4,5] & Chulhong Kim [1,2,3,5,6] ✉

Transparent ultrasound transducers (TUTs) can seamlessly integrate optical and ultrasound components, but acoustic impedance mismatch prohibits existing TUTs from being practical substitutes for conventional opaque ultrasound transducers. Here, we propose a transparent adhesive based on a silicon dioxide-epoxy composite to fabricate matching and backing layers with acoustic impedances of 7.5 and 4–6 MRayl, respectively. By employing these layers, we develop an ultrasensitive, broadband TUT with 63% bandwidth at a single resonance frequency and high optical transparency ( > 80%), comparable to conventional opaque ultrasound transducers. Our TUT maximises both acoustic power and transfer efficiency with maximal spectrum flatness while minimising ringdowns. This enables high contrast and high-definition dual-modal ultrasound and photoacoustic imaging in live animals and humans. Both modalities reach an imaging depth of > 15 mm, with depth-to-resolution ratios exceeding 500 and 370, respectively. This development sets a new standard for TUTs, advancing the possibilities of sensor fusion.

Ultrasound imaging (USI) and optical imaging (OI) sensors are eminently suitable for sensor fusion due to their simplicity, safety, and cost-effectiveness[1]. Employing USI in conjunction with various OI modalities (e.g., photoacoustic imaging (PAI), optical coherence tomography (OCT), fluorescence imaging (FLI), and white light imaging (WLI)) provides complementary information that improves the sensitivity and specificity of disease diagnosis and monitoring while minimizing the drawbacks of each modality alone[2–11]. However, there are two obstacles: 1) seamless integration of USI and OI modalities in a single form factor, and 2) maintaining each modality's best performance. The form factor challenge can be overcome by using a transparent ultrasound transducer (TUT) or a transparent optical detector, which allows for seamless USI and OI integration[3,12,13]. For example, Park et al. demonstrated quadruple imaging modality fusion in a single form factor, and Chen et al. implemented high-speed wide-field optical-resolution PAM (OR-PAM) using a TUT, but their TUT's acoustic performance was inferior to that of a conventional opaque ultrasound transducer (OUT)[3,14]. Ma et al. demonstrated high-quality multi-scale PAI through a transparent, focused optical sensor, but their optical sensor could not generate ultrasound (US) images[15].

There are three prerequisites to fabricating a practical TUT with acoustic performance comparable to that of an OUT: 1) a transparent front matching material with an acoustic impedance of 7–9 MRayl (1 Rayl = 1 Pa·s·m$^{-1}$ = 1 kg/(m$^2$·s)), to maximize transmission efficiency; 2) a

[1]Department of Electrical Engineering, Pohang University of Science and Technology, Pohang, Republic of Korea. [2]Medical Device Innovation Center, Pohang University of Science and Technology, Pohang, Republic of Korea. [3]Department of Convergence IT Engineering, Pohang University of Science and Technology, Pohang, Republic of Korea. [4]Department of Chemistry, Pohang University of Science and Technology, Pohang, Republic of Korea. [5]Department of Medical Science and Engineering, Pohang University of Science and Technology, Pohang, Republic of Korea. [6]Department of Mechanical Engineering, Pohang University of Science and Technology, Pohang, Republic of Korea. [7]These authors contributed equally: Seonghee Cho, Minsu Kim. ✉e-mail: chulhong@postech.edu

transparent backing material of more than 5 MRayl, to eliminate ringdowns by balancing the electrical and acoustic Q factors;[16–18] and 3) a firm connection between all layers, without gaps that degrade the transducer's quality[19–21]. The 1st prerequisite, for the front matching layer, ensures a consistent and high transmit pressure for water especially when utilized with a pure polymer with an acoustic impedance of 2–3 MRayl in a double matching structure. Here, we aim to achieve flat and high gains by utilizing double matching layers consisting of a 7.5 MRayl 1st layer with and a 2.36 MRayl 2nd layer. Detailed information on the front matching is in Supplementary Fig. 1 and Supplementary Note 1. The backing material's acoustic impedance requirement balances the acoustic and electrical Q factors. Given the frontside specifications, backing material with an acoustic impedance of 7.2 MRayl will achieve the optimal bandwidth. In US transducers with similar designs, slightly lighter damping of 5–6 MRayl, is often employed, which does not significantly unbalance the Q factors and enhances sensitivity. This study also aims for lighter damping and accordingly adopts a double-layered backing structure with an acoustic impedance of 6.1 MRayl at the resonance center. The design employs a matching layer of 3.8 MRayl to increase the effective acoustic impedance of the backside without losing transparency. Additional explanations for the backing can be found in Supplementary Note 2. To ensure a solid connection between all layers, the adhesion gap should be minimal compared to the acoustic wavelength. The most effective approach to eliminating the adhesion gap is to use the matching and backing material as an adhesive and directly cure it on the adhesion surface. Hence, this study aims to create a material with a viscosity suitable for adhesive-like bonding. A viscosity of 100 McPs is considered the upper limit because higher-viscosity materials are challenging to pour or spread evenly on a surface.

In this work, to satisfy all three requirements simultaneously, we have used experiments and simulations to devise a recipe that provides the needed acoustic, rheological, and optical properties. Supplementary Fig. 2 illustrates the detailed simulation process. The proposed adhesive $SiO_2$/epoxy composite materials provide optimized optical transparency, acoustic impedance, and flowability. Building on these innovations, we have developed a broadband (63% bandwidth) and ultrasensitive TUT, while maintaining optical

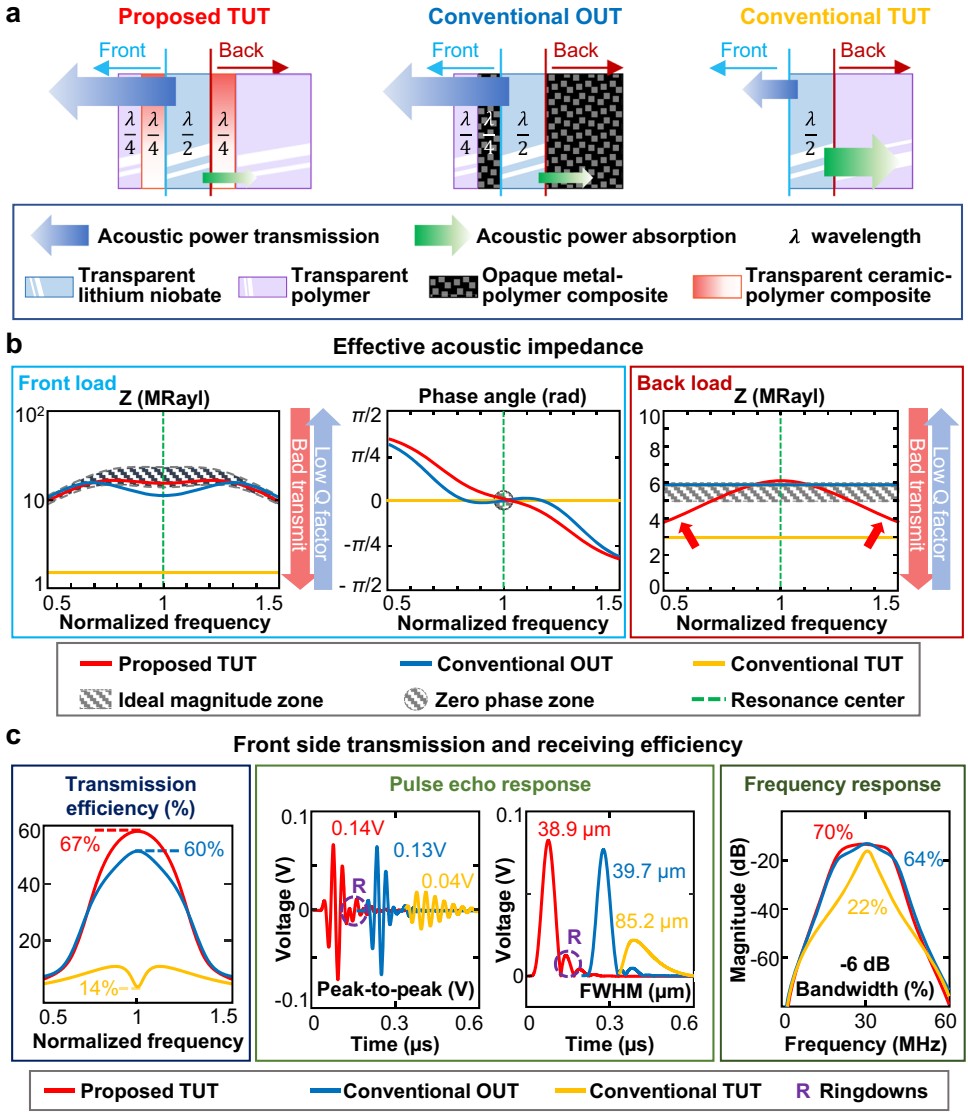

**Fig. 1 | Computed performance comparison of the proposed transparent ultrasound transducer (TUT), a conventional opaque ultrasound transducer (OUT), and a conventional TUT. a** Schematics of the proposed TUT, conventional OUT, and conventional TUT. **b** Comparison of the simulated acoustic impedance magnitudes (MRayl) and phase (rad) at a normalized frequency with a 100% bandwidth. **c** Comparison of the simulated power transmission efficiency and pulse-echo impulse responses. R, ringdown; FWHM, full-width-at-half-maximum; Bandwidth refers to the −6 dB width expressed as a fraction of the center frequency.

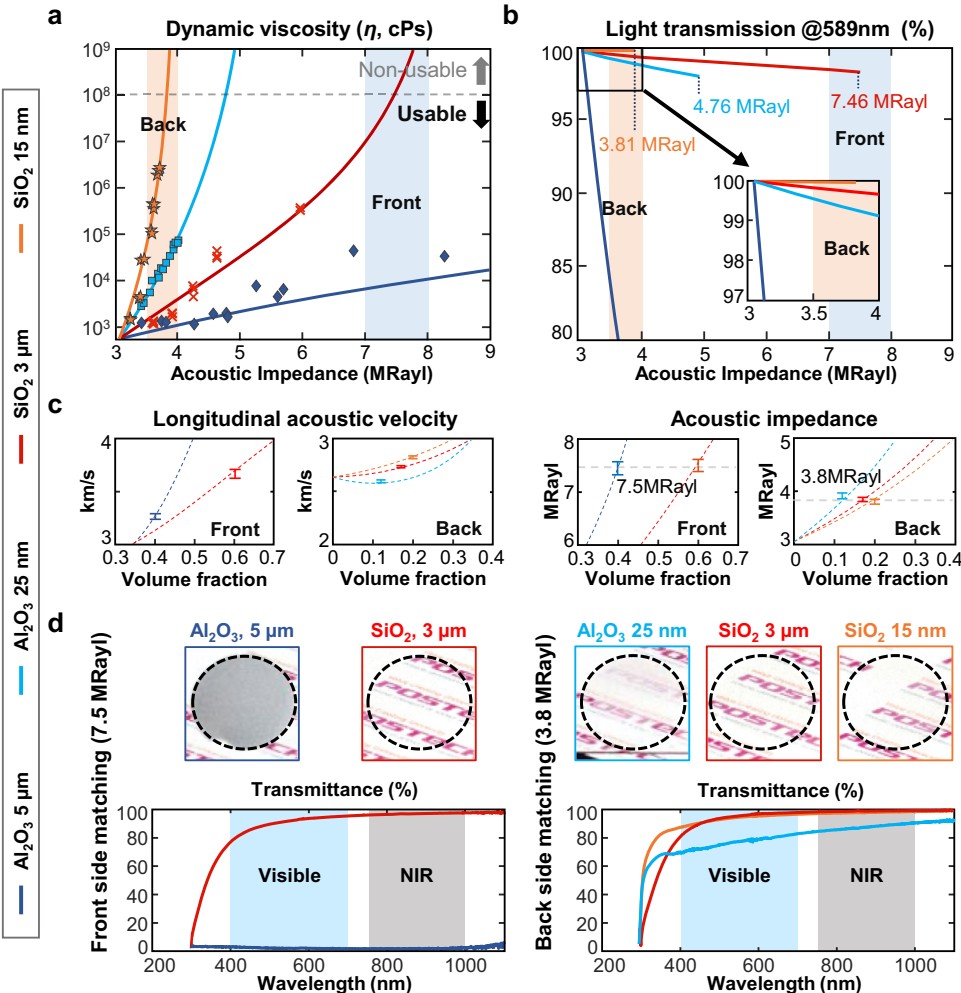

**Fig. 2 | Acoustical, rheological, and optical properties of ceramic-epoxy composites. a** Experimentally measured and theoretically estimated viscosity vs maximum acoustic impedance for various ceramic-epoxy composites. Markers are experimental results, and solid lines are theoretically fitted. **b** Simulated light transmittance at 589 nm of 30 MHz acoustic quarter-wave plates made of various ceramic-epoxy composites. Only simulates up to 100 McPs. **c** Simulated and experimentally measured acoustic impedance and acoustic velocity vs volume fraction for various ceramic-epoxy composites (n = 12). **d** Photographs and experimentally measured optical transmittances of matching layers made of various transparent ceramic-epoxy composites. NIR, near infrared. Error bar denotes standard deviation.

transparency (>80%). Further, we have applied this high-performance TUT in fully integrated dual-modal microscopic USI and PAI in living animals and humans. Using the TUT, the high-definition and high-contrast USI and PAI systems deliver unprecedented performance: an imaging depth of more than 15 mm for both modalities and an axial resolution of 32.6 µm for USI (equivalent to a depth-to-resolution ratio (DRR) of > 500) and 40.4 µm for PAI (equivalent to a DRR of > 370). This significant step forward achieves true dual-modal PAI/USI, providing high-definition USI simultaneously with PAI for the first time. We believe that our research will set a standard for TUT design and advance sensor fusion technologies.

## Results

### Design and simulation of an ultrasensitive and broadband TUT

Figure 1 illustrates the ultrasensitive and broadband TUT's design and compares its computed acoustic performance with those of a conventional OUT[16] and a conventional TUT. Figure 1a shows simple schematics of transducers. In the conventional OUT, metal-epoxy composites are commonly used as both the 1st front matching layer and the backing layer to achieve the desired acoustic impedance and electrical conductivity, thus fulfilling acoustic and viscosity design criteria. Additional detailed information regarding a standard

requirement for a conventional OUT is provided in Supplementary Note 3. To fabricate an ideal TUT, transparent materials must replace opaque components in the conventional OUT. However, transparent composite materials are difficult to produce, so a structure of transparent crystal with a transparent electrode has been investigated[22–25]. Our TUT employs a transparent 7.5 MRayl frontside matching layer, and a transparent 3.8 MRayl backside matching layer increases the effective acoustic impedance to 4–6 MRayl. The two transparent ceramic-epoxy matching layers are fabricated by concurrently optimizing the composite layers' optical scattering and acoustic impedance, as detailed below in the section on "Design and fabrication of a transparent ceramic-epoxy composite material". Lithium niobate (LNO) and indium tin oxide (ITO) are used for the proposed TUT.

The simulated effective acoustic impedance with a 100% fractional bandwidth is shown in Fig. 1b. For successful front pressure loading on a US transducer, the magnitude of the effective acoustic impedance should be in the ideal zone in Fig. 1b (the gray dashed zone, Supplementary Note 1), and the phase angle should indicate a zero-phase characteristic at the center frequency (the gray dashed circle). Figure 1b shows that the acoustic performances of the conventional OUT (solid blue line) and the proposed TUT (solid red line) fall well inside the ideal zones in terms of the front load and phase angle. On

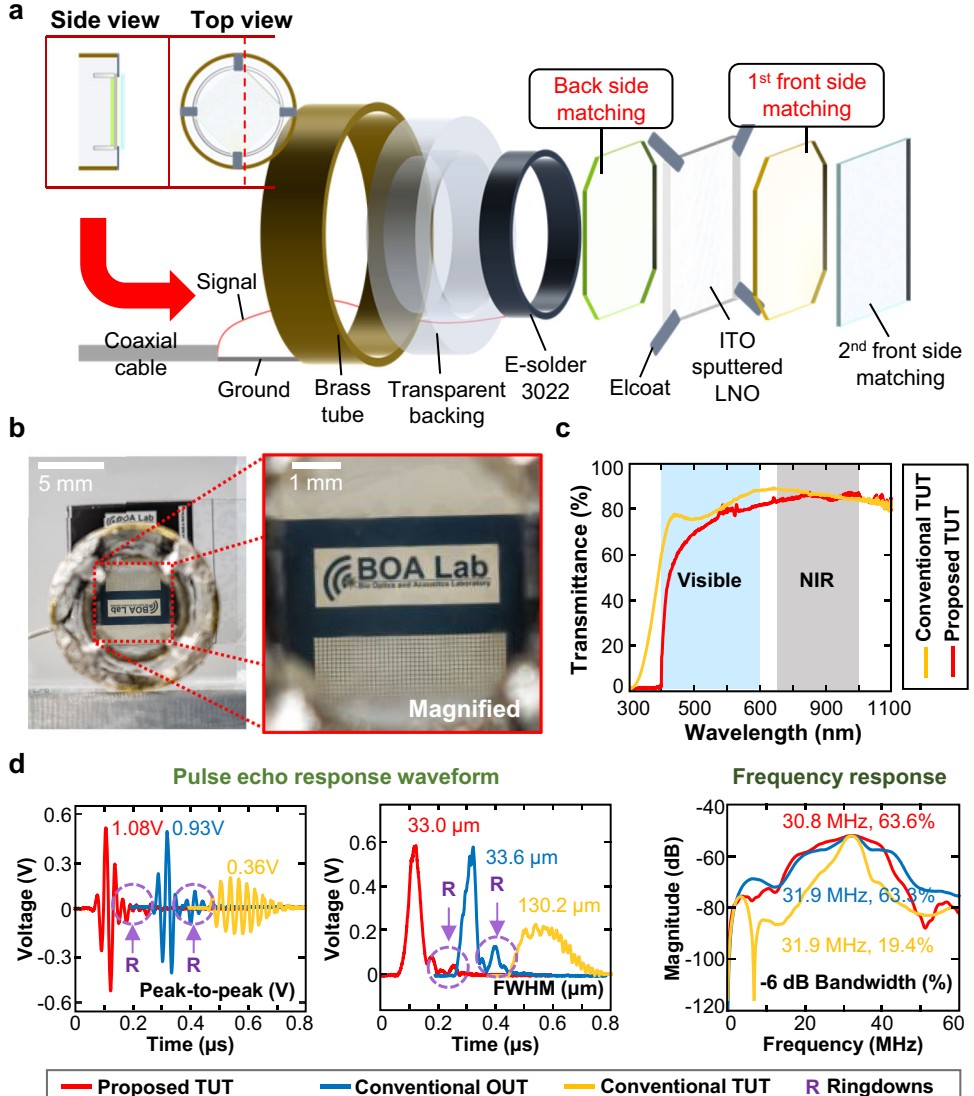

**Fig. 3 | Structure of the proposed transparent ultrasound transducer (TUT) and its optical transmittance. a** Structural schematic of the proposed TUT. **b** Demonstration of the optical clarity of the TUT. **c** Measured optical transmittance of the proposed TUT. **d** Measured pulse-echo responses and spectrum. ITO, indium tin oxide; LNO, lithium niobate; NIR, near infrared; OUT, opaque ultrasound transducer; R, ringdown; FWHM, full-width-at-half-maximum; Bandwidth refers to the -6 dB width expressed as a fraction of the center frequency.

the other hand, the conventional TUT without a front matching layer (solid yellow line) exhibits a significantly low front load, and consequently the transmission efficiency is extremely low, even though the phase angle is in the ideal zone. We also compared the computed acoustic performances of the front loads and phase angles for previously reported TUTs (Supplementary Fig. 3). Note that we incorporate the detailed parameters in[3,19,21,26–28] in the simulations. A detailed comparison can be found in Supplementary Note 4.

For the backside, the effective acoustic impedance magnitude is considered to minimize only the reverberation effects on boundaries, and the phase angle is not considered because a backward echo is not used. An acoustic impedance of 5.9 MRayl in the backside is typical for a conventional OUT, and our proposed TUT exhibits an acoustic impedance of 4–6 MRayl, a comparable value. However, previously reported TUTs with an epoxy backing layer had an acoustic impedance of 3 MRayl, and thus their damping capabilities were quite low, resulting in significant ringdowns.

Figure 1c compares the simulated US transmission capabilities and impulse responses of a conventional OUT (solid blue line), conventional TUT (solid yellow line), and the proposed TUT (solid red line), based on the parameters in Fig. 1b. As shown in the first panel of Fig. 1c, the US transmission efficiency of the proposed TUT at the center frequency is 4.78 times better than that of the conventional TUT. In addition, the pulse-echo responses of the proposed TUT and conventional OUT are very close to each other, except for an additional ringing tail in the proposed TUT ("R" marker in Fig. 1c). This additional ringdown originates from the relatively low effective backload in the peripheral spectral regions (red arrows), but the acoustic performance of the proposed TUT is hardly affected. More importantly, the proposed TUT has a 3-fold higher peak-to-peak voltage amplitude and a 2-fold shorter waveform length than the conventional TUT. Moreover, while the proposed TUT and conventional OUT differ only negligibly in their spectral responses, their bandwidths are significantly wider than that of the conventional TUT. The code used in the pulse echo simulation is provided as Supplementary Code.

## Design and fabrication of transparent ceramic-epoxy composite materials

To increase the acoustic impedance of ceramic-epoxy composite materials while preserving their optical transparency, their optical

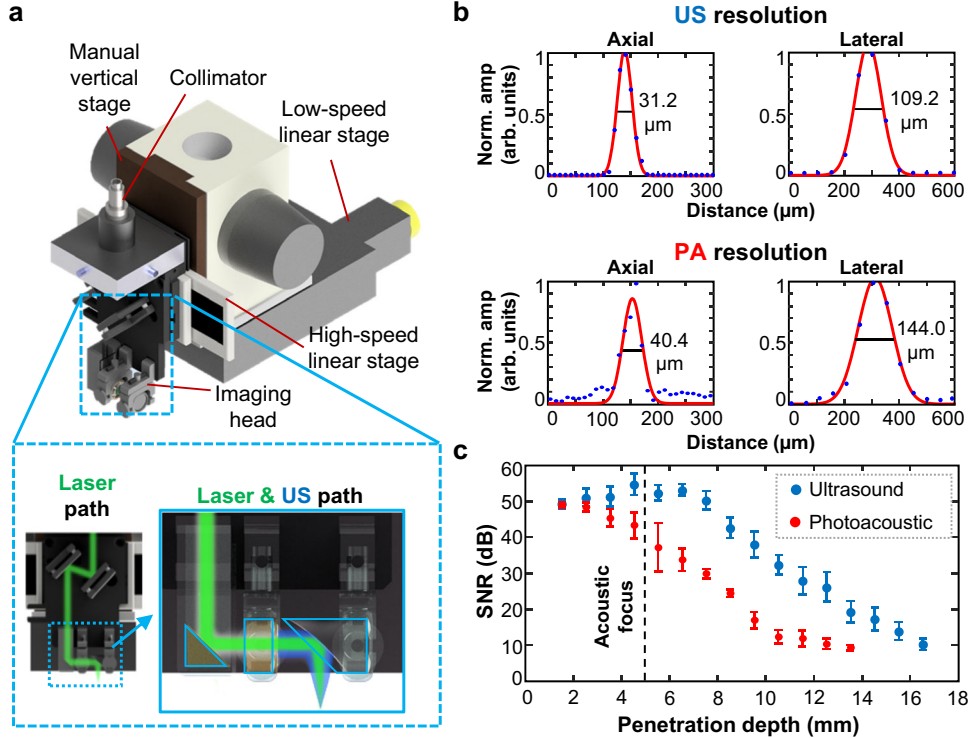

**Fig. 4 | Schematic of an integrated ultrasound (US)/photoacoustic (PA) microscopy system using a high-performance transparent ultrasound transducer (TUT), and its imaging performance. a** An integrated US/PA microscopy system. **b** US/PA full-width-at-half-maximum resolutions. **c** Signal-noise-ratio (SNR) vs penetration depth in chicken breast tissue ($n = 140$). Norm. amp, normalized amplitude; arb. units, arbitrary-units. Error bar denotes standard deviation.

scattering, volume fraction, and viscosity must be simultaneously controlled[29,30]. Figure 2 shows acoustic, rheological, and optical analyses of several ceramic-epoxy composites with a usable viscosity limit. The measured dynamic viscosities for the acoustic impedances of the various composites are marked in Fig. 2a, and the results are theoretically fitted (solid lines).

Figure 2b shows the simulated light transmittance of 30 MHz quarter-wave plates as a function of the acoustic impedance of the ceramic-epoxy composite materials in Fig. 2a. The results in Fig. 2a and b indicate that composites made with $SiO_2$ micro-/nano-powders and $Al_2O_3$ nano-powders (red, orange, and light blue lines, respectively) are both practical for the backside matching layer. However, $SiO_2$ micro-powder composites (red line) are the only acceptable candidate for the front matching layer, and $Al_2O_3$ micro-powder composites (dark blue line) are not suitable for either side. All detailed parameters used in the calculations are listed in Supplementary Table 1.

Figure 2c shows the simulated longitudinal acoustic velocity and acoustic impedance of each composite vs its volume fraction and measured values. Figure 2d shows photographs and the experimentally measured optical transmittances of matching layers made of various transparent ceramic-epoxy composites. The matching layer sample made of $SiO_2$ micro-powder-epoxy composite exhibits the best acoustic impedance for frontside matching (7.5 MRayl) with high transparency, while the $Al_2O_3$ micro powder composite is opaque. Among the three candidates for the backside matching, the $SiO_2$ micro- and nano-power epoxy composites are highly transparent, whereas the $Al_2O_3$ nano-powder one is not. Based on these results, the $SiO_2$ micro-powder composite material, with an acoustic impedance of 7.5 MRayl, was selected for the first frontside matching layer, and the same composite formulated with an impedance of 3.8 MRayl was selected for the backside matching layer. The selected front and backside matching layers exhibited over 95 and 98% transmittance in the NIR region and over 90% and 94% in the visible region, respectively.

## Fabricating the transparent ultrasound transducer and benchmarking its optical and acoustic performance

Using the selected matching layers shown above, we fabricated highly sensitive, broadband, and single resonance TUTs. Figure 3a shows the detailed structure of an example TUT, and the acoustic parameters of all components are summarized in Supplementary Table 2. Basically, a transducer stack consisting of a square, ITO-sputtered LNO crystal contained between matching layers is connected to a coaxial wire, and then is secured in a brass housing using a transparent backing material. All the transducers in this study used square LNO crystals (5.6 mm × 5.6 mm, and 100 μm thick), with quarter-wavelength matching layers. Figure 3b a macro photograph, shows the optical clarity of the fabricated TUT, clearly revealing a 100 μm spacing. Figure 3c displays the optical transmittances of the proposed TUT and a conventional TUT, showing a slightly lower transmittance for the proposed TUT in the visible region, but nearly identical transmittances for both TUTs in the NIR range.

Figure 3d shows the comparative US pulse-echo responses of a conventional OUT (solid blue line), a conventional TUT (solid yellow line), and the proposed TUT (solid red line). The pulse-echo responses of the proposed TUT and conventional OUT are similar. Both TUTs have an additional ringing tail in the echo response (labeled "R" in Fig. 3d). More importantly, compared to the conventional TUT, the proposed TUT has a 3-fold higher peak-to-peak voltage amplitude and a 3.9-fold shorter waveform length. Moreover, the proposed TUT and conventional OUT exhibit negligible differences in their spectral responses, but their bandwidths are significantly wider than that of the conventional TUT. The experimental measurements in Fig. 3d are very close to the simulated results in Fig. 1c. Supplementary Fig. 4 shows the simulated and experimentally measured electric input impedance spectra of all three types of transducers, proving that our proposed TUT can successfully generate a single broadband resonance with minimal inter-layer defects. The experimental and simulation results

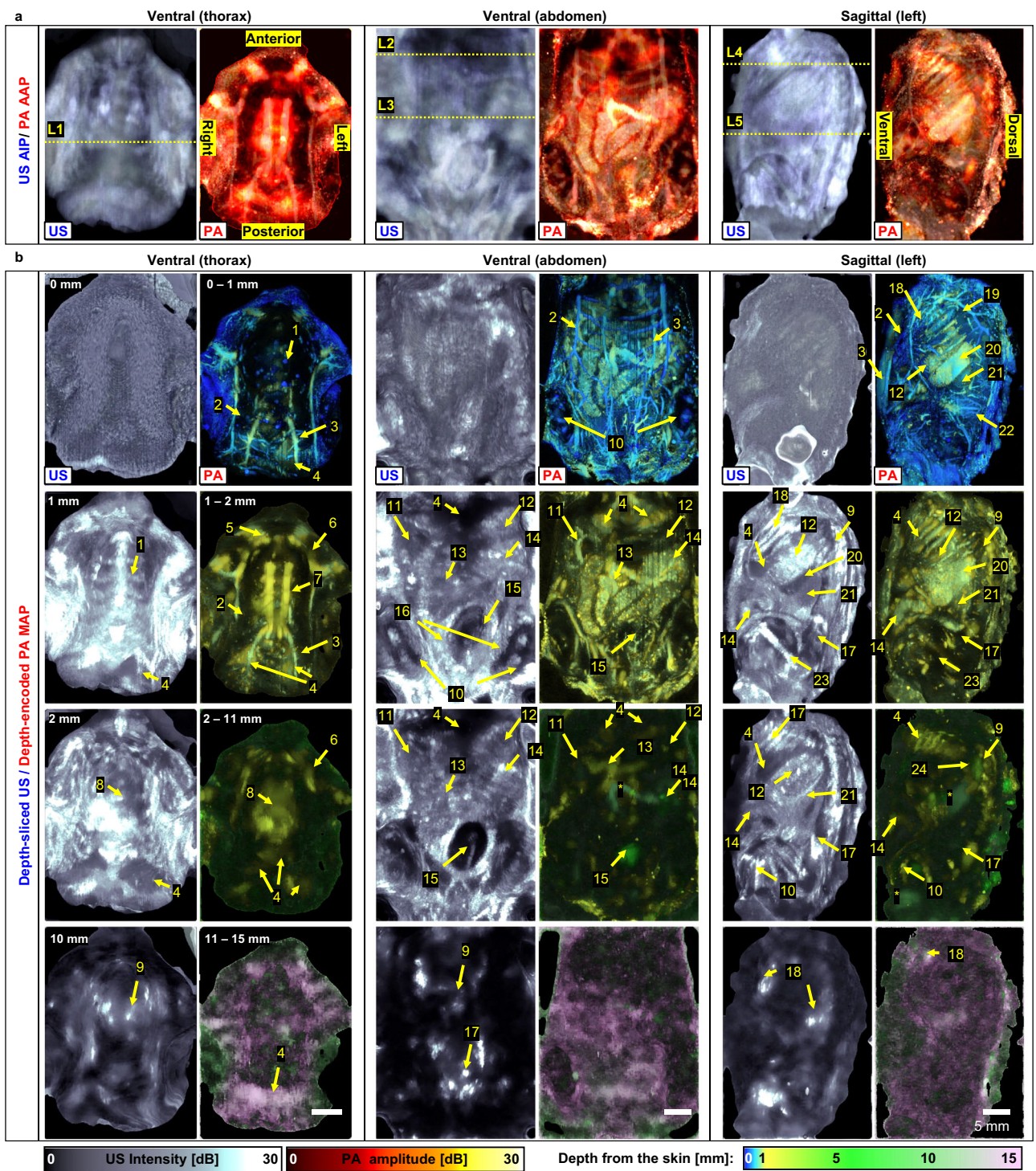

**Fig. 5 | Volumetric ultrasound (US) and photoacoustic (PA) images of a live mouse. a** US average intensity projection (AIP) and PA average amplitude projection (AAP) images of the mouse in the ventral (thorax and abdomen) and sagittal planes. **b** Depth-sliced US and depth-encoded PA maximum amplitude projection (MAP) images at various depths. 1, sternum; 2, mammary vessel; 3, epigastric vessel; 4, liver; 5, common carotid vessel; 6, subclavian vessel; 7, internal thoracic vessel; 8, heart; 9, vertebra; 10, tibia; 11, duodenum; 12, stomach; 13, jejunum; 14, cecum; 15, bladder; 16, preputial gland; 17, pelvis; 18, rib; 19, lateral thoracic vessel; 20, spleen; 21, kidney; 22, circumflex iliac vessel; 23, femur; 24, scapula; *, reverberation artifact.

confirm that, acoustically and electrically, the proposed TUT performs almost identically to the conventional OUT.

## Combined ultrasound/photoacoustic microscopy system using the TUT, and its imaging results in live animals and humans

To investigate the potential of seamlessly integrated US/photoacoustic (PA) imaging using the ultrasensitive TUT, we benchmarked the system's performance in phantoms and chicken breast tissue. A schematic of the combined US/PA microscope is shown in Fig. 4a, with plots of its imaging performance. Our system is specifically designed for deep tissue imaging using acoustic-resolution photoacoustic microscopy. The integrated USI/PAI system consists of an imaging head and a raster scanning mechanism with two horizontal linear stages and one vertical stage. The imaging head consists of two

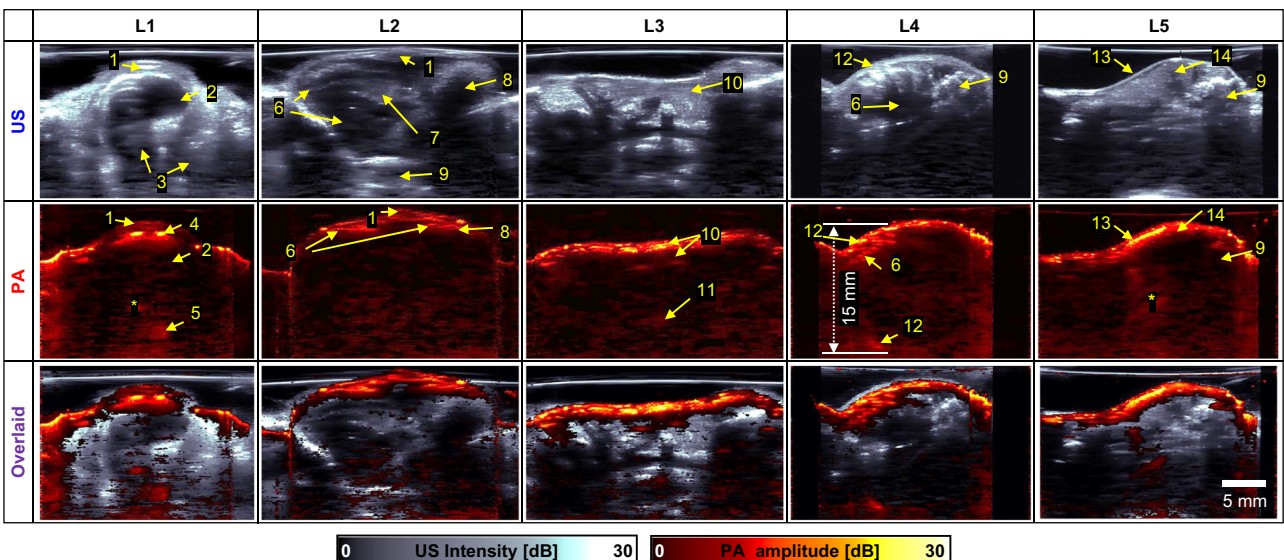

**Fig. 6 | 2D cross-sectional ultrasound (US)/ photoacoustic (PA) images cut along lines L1-5 in Fig. 5.** 1, sternum; 2, heart; 3, lung; 4, internal thoracic vessels; 5, thoracic vessel; 6, liver; 7, gallbladder; 8, stomach; 9, vertebra; 10, jejunum; 11, abdominal vessel; 12, rib; 13, spleen; 14, kidney; *, reverberation artifact.

angle-adjustable mirrors and a fixed right-angle mirror prism, which direct the output of a 1064 nm laser through the TUT. A parabolic mirror simultaneously focuses both the laser beam, after it passes through the TUT, and the US beam. In USI mode, the acoustic waves generated by the TUT are reflected by the mirror into the sample, and then the generated backscattered waves are received by the TUT. In PAI mode, the laser beam passing through the TUT is reflected by the mirror and irradiates the sample. The generated PA waves are then detected by the TUT. The axial resolutions for USI and PAI are $32.6 \pm 1.9$ μm and $40.4 \pm 1.6$ μm, respectively, and the lateral resolutions are $110.4 \pm 21.8$ μm and $148.4 \pm 13.5$ μm, respectively (Fig. 4b). The measured axial and lateral resolutions show a strong correlation with theoretical values. Detailed comparisons and information for calculating theoretical values are presented in Supplementary Table 3 and Supplementary Fig. 5. Figure 4c and Supplementary Fig. 6 show the signal-to-noise ratio (SNR) according to the depth. The SNR for USI initially increases until the acoustic focal spot is reached, and then it decreases, but the SNR for PAI decreases from the beginning, due to optical attenuation. The highest SNRs for USI and PAI are 55 and 49 dB, respectively. The maximum penetration depths of US and PA in chicken breast tissue are 16.6 mm and 13.6 mm, respectively.

To investigate the potential of in-vivo high-resolution and high-contrast USI/PAI with the ultrasensitive TUT, we imaged a live mouse (Fig. 5 and Fig. 6). Figure 5 displays US and PA images of the mouse's body in ventral (thorax and abdominal regions) and sagittal (from the left side of the body) views. Figure 5a shows US average intensity projection (AIP) and PA average amplitude projection (AAP) images from the imaging planes. Figure 5b shows depth-sliced US and depth-encoded PA maximum amplitude projection (MAP) images at various depths. Note that depth-sliced US images are used instead of the local US maximum intensity projection (MIP) images because the background speckle patterns are too dominant in the local US MIP images. Depth-sectioning movies are provided in Supplementary Movies 1–3. The US AIP images show the overall anatomy of the mouse's body, such as bones, soft tissues, and organs. The major blood vessels, heart, and liver in the thorax region are clearly visible in the ventral PA AAP image. In the abdominal region ventral PA AAP image, the gastrointestinal tract, bladder, and superficial vessels are clearly visible. Furthermore, the sagittal PA AAP image clearly shows the kidney, spleen, stomach, liver, cecum, vertebra, rib, pelvis, femur, scapula, and blood vessels.

Details are revealed in the depth-sliced US and PA images, as illustrated in Supplementary Note 5.

Figure 6 shows cross-sectional US, PA, and overlaid US/PA B-mode images cut along lines 1–5 (L1–5) in Fig. 6. The morphological US images are complementary to the PA images based on optical absorption. The B-mode images show different regions, including the mid-thorax (L1), upper abdomen (L2), the middle section of the abdomen (L3), the bottom of the thorax from the left side (L4), and from left side of the lumbar region (L5). Interestingly, the top and bottom boundaries of the thoracic cavity are both captured in the PA image, and the imaging depth reaches up to 15 mm below the skin surface at L4. Details are described in Supplementary Note 6.

In a more clinically relevant demonstration, we performed in-vivo high-definition imaging of a human palm with the ultrasensitive TUT (Fig. 7. and Fig. 8). Imaging blood vessels in human extremities are important in various diseases such as diabetes and Raynaud's diseases. Figure 7a shows US AIP and PA AAP images of the human palm, and Fig. 7b shows various depth-sliced US and depth-encoded PA MAP images. A depth-sectioning movie of the human palm is provided in Supplementary Movie 4. The US AIP image shows the overall tissue structure, including the skin, subcutaneous tissue, muscles, and blood vessels. Vasculature features (e.g., subcutaneous vessels, palmar veins, and arteries) are particularly easy to see in the PA AAP image. The depth-sliced US and PA images show the following details: (1) the fine handprint texture at the surface of the skin, (2) subcutaneous vessels below 0.6 mm deep, (3) palmar veins at 1.4–4 mm deep, and (4) arteries at over 4 mm deep.

Figure 8 shows cross-sectional US, PA, and overlaid US/PA B-mode images cut along the yellow dotted line in Fig. 7a US. The B-mode US image displays the palmar tissue as discrete layers (e.g., epidermis, dermis, subcutaneous tissue, fat pad, and muscles), such as the epidermal junction (red-dashed line), the dermal-subcutaneous junction (yellow-dashed line), and palmar aponeurosis (purple-dashed line). The palmar veins and arteries are visible in the subcutaneous tissue and fat pad below the aponeurosis, respectively. Moreover, the adductor pollicis muscle is visible in the the near region below the subcutaneous tissue. The B-mode PA image displays the epidermis and blood vessels (e.g., subcutaneous vessels, palmar veins, and arteries), and the imaging depth reaches to 6.5 mm below the skin.

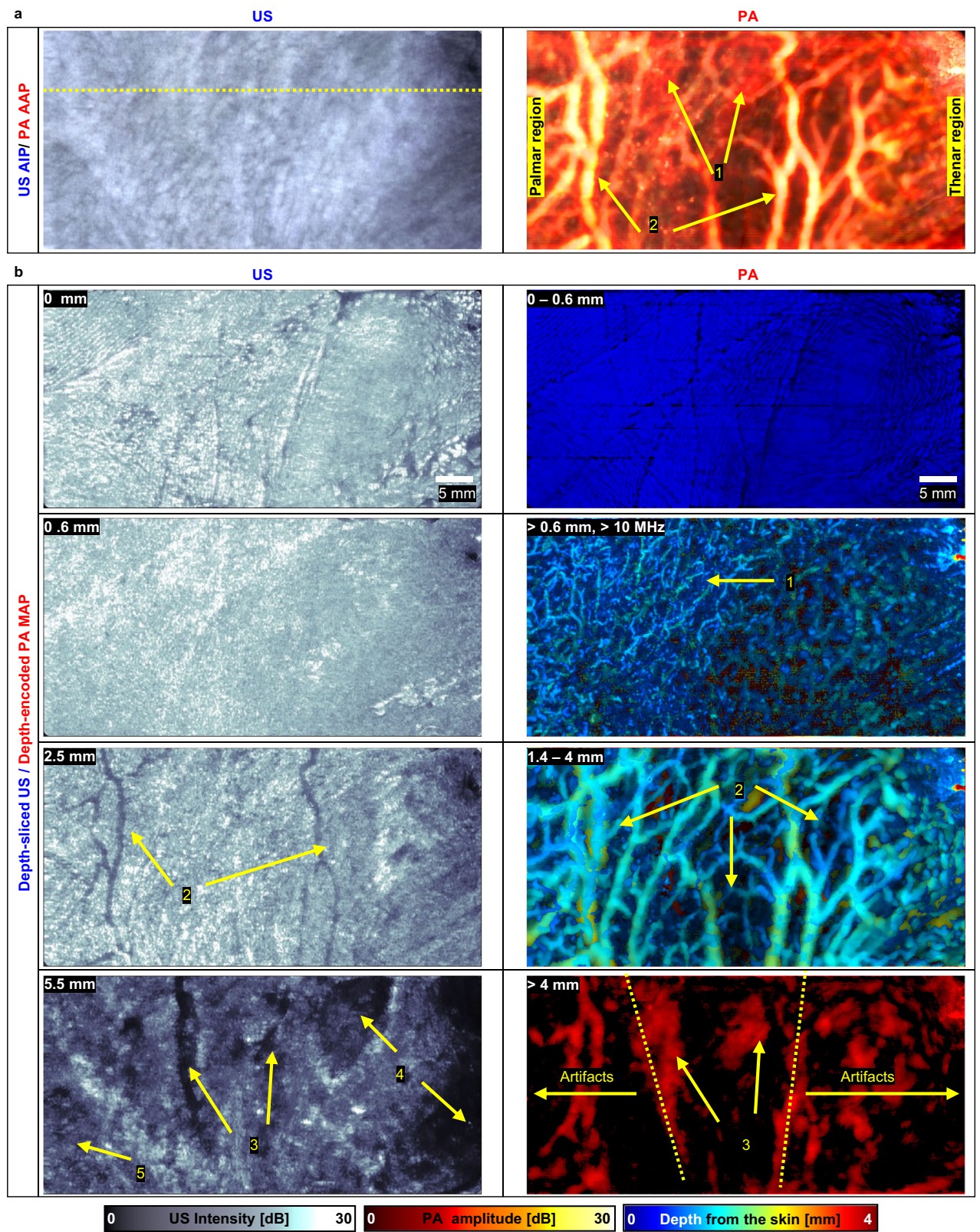

**Fig. 7 | Volumetric ultrasound (US) and photoacoustic (PA) images of a human palm. a** US average intensity projection (AIP) and PA average amplitude projection (AAP) images of a human palm. **b** Depth-sliced US and depth-encoded PA maximum amplitude projection (MAP) images at various depths. 1, sub-cutaneous vessels; 2, palmar veins; 3, palmar digital arteries; 4, adductor pollicis muscles; 5, adductor minimi muscle.

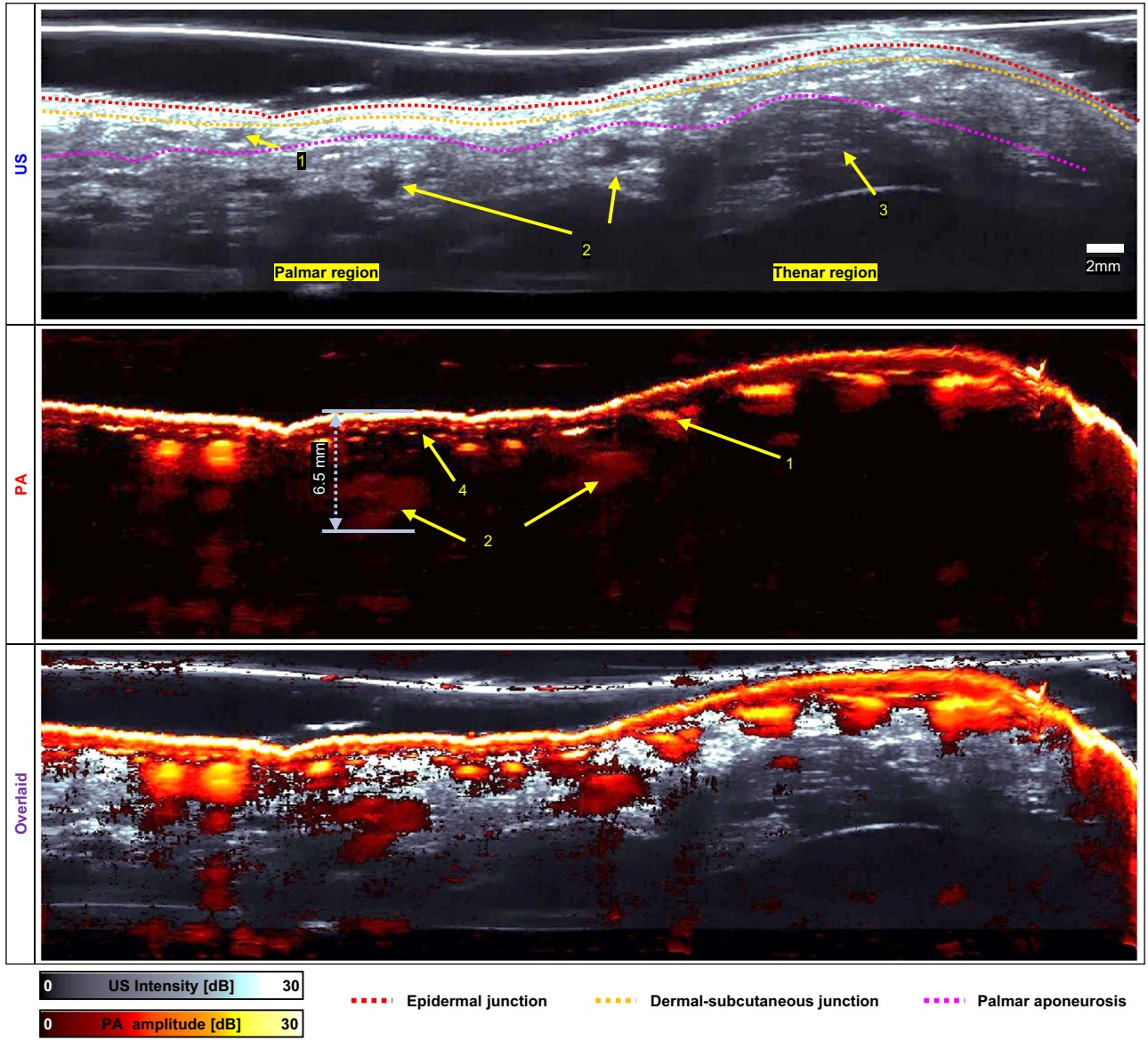

**Fig. 8 | 2D cross-sectional ultrasound (US)/ photoacoustic (PA) images cut along the line in Fig. 7a US.** 1, palmar vein; 2, common palmar digital artery; 3, adductor pollicis; 4, subcutaneous vessels.

## In-vivo whole-body perfusion kinematics monitoring of infrared dye

The proposed TUT imaging system was utilized to monitor the perfusion of an infrared dye. A time-series of PA images was acquired before and after intravenous injection of IR-1048 dye solution for 9 hours (Fig. 9). Figure 9a shows ventral-view PA MAP images of the entire mouse body pre- and post-injection. A rapid signal increase is observed in the liver region at 30 mins postinjection, and then the signal gradually decreases over time, ultimately returning to its initial brightness after 6 hours. This behavior suggests that the liver quickly takes up the injected dye, and the dye then gradually clears out. The magnified areas in the blue boxes showcase an intensified signal within the liver, characterized by a distinctive granular texture that could result from dye uptake within the hepatic lobules. Unlike the liver, the bladder displays a signal increase at 3 h and 6 h postinjection, returning to its initial state by 9 h postinjection. It can be inferred that the dye is quickly taken up by the liver and then slowly cleared out via the bladder. The 30-min and 3-h images also show numerous small bright signal dots throughout the whole body, which could indicate that the dye is excreted through the skin. Figure 9b and c are cross-sectional PA

images of the liver and bladder regions over time, respectively. The PA signal peaks in the liver at 30-min postinjection, and then decreases. This behavior differs from that of the bladder signal, which peaks at 3 hr post-injection and then decreases. Figure 9d illustrates the changes in the PA signal over time. In the liver, the PA signal is dramatically enhanced at 30 min after injection (5.58 ± 0.08, arb. units), and it still displays an enhanced signal at the 3 h mark (3.70 ± 0.18, arb. units). However, by 6 h, it has returned to an amplitude similar to the control (1.03 ± 0.19, arb. units). In the bladder, the signal gradually increases up to 3 h postinjection (1.98 ± 0.10, arb. units) and then has returned to its initial level by 9 h postinjection (1.15 ± 0.13, arb. units).

Our in-vivo imaging results were validated through ex-vivo studies. Figure 9e shows photographs and PA MAP images of excised liver samples with and without dye injection. The dyed liver was excised at 30 min post-injection, and the control liver was removed from a healthy mouse. Although both excised livers appear similar in the photograph, the PA signals acquired from the excised liver with dye injection are significantly stronger than those from the control liver. When we quantified the average signal difference between ex-vivo samples (Fig. 9f), liver samples from dye-injected mice showed an

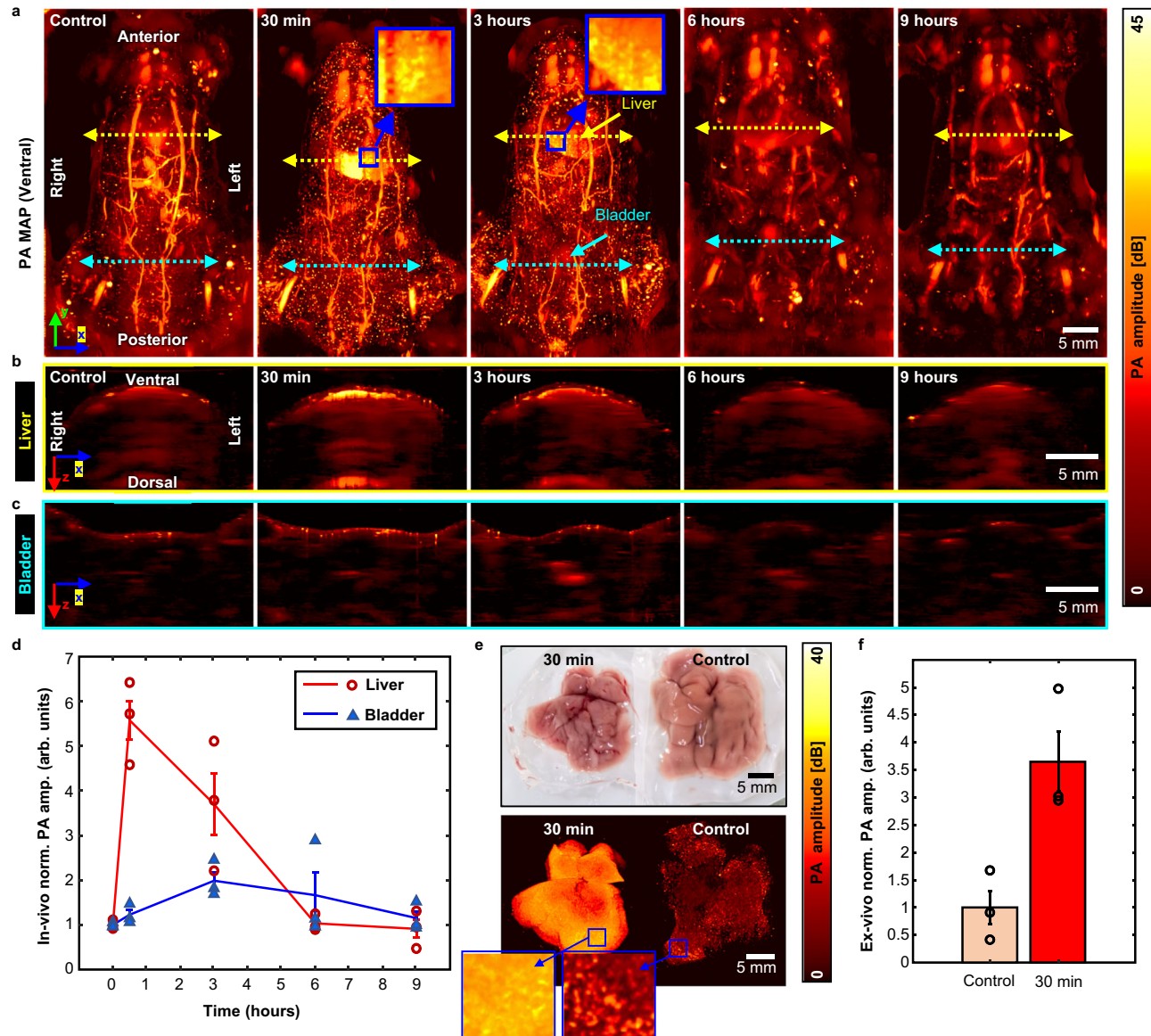

**Fig. 9 | Perfusion kinematics monitoring of IR-1048 dye in live animals.**
**a** Photoacoustic (PA) maximum amplitude projection (MAP) images of a live mouse from before injection to 9 hours after injection. **b** 2D cross-sectional PA images cut along the line on the liver. **c** 2D cross-sectional PA images cut along the line on the bladder. **d** In-vivo normalized PA amplitude profile in the liver and bladder regions over time after injection ($n = 3$). **e** Ex-vivo validation of IR-1048 accumulation in the liver at 30 minutes after injection. **f** Ex-vivo normalized PA amplitude profile in the extracted liver ($n = 3$). Norm. amp, normalized amplitude; arb. units, arbitrary-units. Error bar denotes standard error of three experiments.

approximately 3.7-fold enhanced signal ($3.65 \pm 0.54$, arb. units) over that of control livers ($1.00 \pm 0.30$, arb. units).

## Discussion

We designed and built an ultrasensitive, broadband, and single resonance TUT by introducing proposed transparent thin-film ceramic-polymer matching layers on the piezo crystal's front and back sides. We fabricated an optically transparent 7.5 MRayl matching layer (>95%) and 4–6 MRayl backing (>98%). In a US pulse-echo test, our proposed TUT demonstrated a 63% bandwidth with a 30 MHz single resonance. It offers pulse-echo sensitivity identical to that of a conventional OUT, with high optical transparency (>80%). As detailed in Supplementary Fig. 7 and Supplementary Table 4, the performance of our proposed TUT, including its efficiency and noise-equivalent-pressure (NEP)[31], is compared with those of commercial transducers commonly used for the AR-PAM, a custom-made conventional TUT, a custom-made OUT, and a custom-made ring-shaped ultrasound

transducer (RUT). The custom-made US transducers exhibit higher efficiency than the commercial ones. Among the custom-made US transducers, the NEP/$\Delta f$ is very comparable between the OUT, RUT, and proposed TUT. In terms of optimal coaxial alignment between the optical and acoustical beams, the custom-made RUT and proposed TUT are superior. However, the sidelobes of the acoustic fields are relatively severe in the RUT, and consequently, the lateral resolution of the RUT is relatively worse than that of the proposed TUT. The results of simulations and experiments to compare the lateral resolutions of the custom-made RUT and proposed TUT are summarized in Supplementary Table 5 and Supplementary Figs. 8 and 9. While optical detectors such as micro-ring resonators are very sensitive in detecting acoustic signals, and they are suitable for small form factors, they are almost impossible to generate US pressure[32]. Although the optical detectors are good candidates for PA imaging, they are not best suited for dual-modal PA/US imaging. Using our proposed TUT, we demonstrated high-performance USI/PAI in-vivo, achieving axial resolutions

of 32.6 μm and 40.4 μm for USI and PAI, respectively, and lateral resolutions of 110.4 μm and 148.4 μm, for USI and PAI, respectively. In chicken breast tissue, USI and PAI exhibited 55 dB and 49 dB SNRs, respectively. Regarding the quality of acoustic resolution imaging, although we employed a parabolic mirror, which can limit the system's performance by constraining the acoustic numerical aperture (NA$_A$) compared to a concave lens, our current imaging system outperforms our previous design[3] by providing a greater imaging range while preserving a comparable lateral resolution. This enhancement is demonstrated in Supplementary Fig. 10. The maximum imaging depth varied, depending on the sample. In chicken breast tissue, USI and PAI achieved imaging depths of 16.6 mm (a DRR of 509) and 13.6 mm (a DRR of 337), respectively. The mouse body exhibited low acoustic wave attenuation, enabling PAI depths of up to 15 mm to be reached (a DRR of 371). However, the human hand showed strong acoustic wave attenuation, resulting in relatively shallow USI and PAI penetration depths of 9.0 mm (a DRR of 276) and 6.5 mm (a DRR of 161), respectively. The DRRs presented here are the best performances among reported studies on microscopic PAI (Supplementary Fig. 11 and Supplementary Table 6). Compared to previously reported studies, our work shows significant in-vivo imaging improvements. Our study shows a 3–6 fold (371 vs 56–120) DRR enhancement in live mouse imaging and about a 3-fold (163 vs 40–43) DDR enhancement in human palm imaging. In addition, this is the only study that simultaneously provides high-resolution USI.

Compared to previous AR-PAM studies, the high-DRR system in this study offers enhanced performance, providing deeper penetration and larger in-vivo FOV images. The proposed TUT employed here exhibited higher detection efficiency and lower NEP than the commercial transducers used in previously reported systems (Supplementary Fig. 3 and Supplementary Table 4). The proposed TUT also works as a transparent optical window. These innovations combine to offer significantly expanded imaging capabilities. While most previous studies primarily visualized blood vessels within superficial layers in mice, such as the peritoneum and skin, reaching no deeper than 3 mm below the skin surface[33–37], our system can visualize even common organ structures in mice. In a previous study, an imaging system utilizing dark-field illumination and a 5 MHz frequency transducer could penetrate deeply enough to visualize common organ structures in mice, but it sacrificed resolution to do so[38]. Moreover, its penetration depth was inferior to that of our system (10.3 mm vs. 15 mm). Still another dark-field illumination approach, using a 30 MHz transducer and a 1064 nm wavelength, demonstrated shallower penetration into a chicken breast phantom (11 mm vs. 13 mm) and comparable spatial resolution to our system (57 μm vs. 40 μm axially, 130 μm vs. 148 μm laterally), using about 8 times higher laser energy (1.8 mJ vs. 220 μJ)[39]. Its phantom experiment using a black-tape target demonstrated relatively deep penetration compared to other studies. However, it provided images of in-vivo black ink injections only at limited depth, which presented challenges in visualizing signals with an insufficient SNR from deeper mouse tissue components. In contrast, we visualized not only the PA signal from blood vessels but also the normal tissue of mice, as demonstrated in the B-scan images presented in Figs. 6 and 9. In the case of human palm imaging, while other studies primarily visualized shallow blood vessels within 2 mm depth[33,40], our study successfully distinguished and visualized the skin, subcutaneous vessels, veins, and arteries.

The improved sensitivity of the proposed TUT not only significantly improved the DRR of the images but also properly adjusted the NA$_A$, as discussed further in Supplementary Note 7. In summary, we believe that our proposed TUT and its experimental fabrication procedure open a new direction for the practical implementation of TUTs and accelerate the seamless fusion of optical and acoustic sensors for applications in medical research and industry.

## Methods

### Analysis of the effects of matching and backing layers on ultrasound transducers

To analyze the acoustic energy transfer and damping characteristics of an ultrasound (US) transducer, the acoustic load impedance is calculated. In this work, all calculations were based on one-dimensional acoustic transmission line simulations. The effective acoustic impedance of a serial acoustic transmission line is defined by

$$Z_{input} = Z_1 \frac{Z_2 + iZ_1 \tan(\frac{2\pi}{\lambda} l)}{Z_1 + iZ_2 \tan(\frac{2\pi}{\lambda} l)}, \qquad (1)$$

where $Z_{input}$ is the acoustic input impedance, $Z_1$ is the transmission line's acoustic impedance, $Z_2$ is the loading's acoustic impedance, $\lambda$ is the acoustic wavelength, and $l$ is the transmission line's length. Here, the power transmission efficiency was calculated based on a three-port acoustic transmission line network. The transducers described in this manuscript were simulated by the Krimholtz, Leedom, and Matthae (KLM) model[41] to analyze their pulse-echo performance and the electrical input impedance. A custom MATLAB-based KLM model simulator using the ABCD matrix computation scheme[42] was built and used. The pulse-echo waveforms and response spectra were generated for a perfect reflection target, and simulation simulations did not consider signal loss in the propagation medium. We determined that the acoustic impedances of the matching materials for the proposed TUT were similar to those of a conventional OUT[16]. In the simulation, the proposed TUT was considered to use a double quarter-wave matching layer (7.5 MRayl and 2.4 MRayl) on the frontside and a single quarter-wave matching layer (3.8 MRayl) on the backside. The impedance of the backing layer is 2.4 MRayl.

### Acoustical simulation of the ceramic-epoxy composites and their experimental validation

The longitudinal acoustic velocity, $C_L$, of the cured ceramic-epoxy composite resin was determined for use in the KLM simulation and the calculation of acoustic impedance. The value of $C_L$ was calculated using the bulk modulus $K$, shear modulus $G$, and density $\rho$, per the following the equation:

$$C_L = \sqrt{\frac{K + \frac{4}{3} G}{\rho}}. \qquad (2)$$

For the ceramic-epoxy 0-3 composite, $K$ and $G$ are given by the Devaney model:[43,44]

$$K = K_{matrix} + V \frac{(3K + 4G)(K_{filler} - K_{matrix})}{3K + 4G + 3(K_{filler} - K_{matrix})}, \qquad (3)$$

$$G = G_{matrix} + V \frac{5(3K + 4G)G(G_{filler} - G_{matrix})}{(15K + 20G)G + 6(K + 2G)(G_{filler} - G_{matrix})}, \qquad (4)$$

where $K_{matrix}$, $G_{matrix}$, $K_{filler}$, and $G_{filler}$ are the matrix and filler particles's bulk modulus and shear modulus and $V$ is the volume fraction of the filler particles. The density, $\rho$, of a ceramic-epoxy composite is given by

$$\rho = (1 - V)\rho_{matrix} + V\rho_{filler}, \qquad (5)$$

where $\rho_{matrix}$ and $\rho_{filler}$ are the densities of the epoxy matrix and ceramic particle filler. Because the acoustic impedance, $Z$, is defined by product of the longitudinal acoustic velocity and density, the

impedance is represented by

$$Z = \sqrt{\left(K + \frac{4}{3}G\right)\left((1-V)\rho_{matrix} + V\rho_{filler}\right)}. \tag{6}$$

The simulated longitudinal acoustic velocity was validated by measuring the US travel time in a sample. Circular chip samples of ceramic-epoxy composites with a thickness between 0.7–2 mm was fabricated for the experimental measurements, and twelve samples were measured in each case. A US transducer (V212-BC-RM, Olympus-NDT, USA), a pulse receiver (DPR 500, Imaginant Inc. USA), and an oscilloscope (MSO 5204, Tektronix, USA) were used to measure the US round trip time, and a height gauge (Messstativ CS 200, HEIDENHAIN, Germany) measured the chip's thickness.

We calculated the acoustic impedance of the material by measuring the travel time of acoustic waves and obtaining the weight and volume of a half-inch diameter chip with a thickness between 0.7–2 mm. The dimensions of the sample chips were controlled to ±1 μm. The density of each sample was calculated using the measured volume and weight values, and we multiplied this value by the longitudinal velocity of each sample.

## Rheological analysis of the ceramic-epoxy composites

The dynamic viscosity of the uncured ceramic-epoxy composite slurry was estimated as a function of the particle volume fraction by fitting the measured dynamic viscosity data to the Krieger and Dougherty model[45]. To avoid viscosity change due to curing, the dynamic viscosity of the ceramic-epoxy composite was measured in the slurry paste state without a hardener. A rheometer (HR20, TA Instruments, USA) was used in an oscillating mode at 25 °C, with a 10 rad/s angular frequency and a 2% oscillation displacement. Based on the dynamic viscosity data, the intrinsic viscosity of the ceramic filler particles was estimated by the following equation:[46]

$$[\eta] = \lim_{V \to 0} \frac{\eta - \eta_0}{\eta_0 V} \approx \frac{\eta - \eta_0}{\eta_0 V} \ (for \ diluted \ solution), \tag{7}$$

where $[\eta]$ is the intrinsic viscosity of the ceramic filler particles and $\eta_O$ is the dynamic viscosity of the pure epoxy matrix. For the intrinsic viscosity estimation, five samples of dynamic viscosity data with $V$ less than 0.05 were employed. The maximum volume fraction, $V_{max}$, and the dynamic viscosity function according to the volume fraction of ceramic particles were estimated by the Krieger and Dougherty model[45] as follows:

$$\eta = \eta_0 \left(1 - \frac{V}{V_{max}}\right)^{-[\eta] \cdot V_{max}}. \tag{8}$$

To estimate the maximum volume fraction and the dynamic viscosity function, 15 samples with a $V$ value greater than 0.05 were used. All estimations were performed using the least-squares method.

## Light transmission simulation and measurement

Light transmission values were obtained from a scattering simulation with Mie theory. Because negligible light is absorbed by the ceramic particles and transparent epoxy, the light transmission (%) is given by

$$Light \ transmission(\%) = 100e^{-\mu_s' d}, \tag{9}$$

where $\mu_s'$ is the reduced scattering coefficient and $d$ is the thickness of the cured ceramic-epoxy composite. The light transmittance of the composite material was measured for a matching layer thickness circular chip sample mounted on a slide glass, using a UV-VIS spectrophotometer (S-3100, SCINCO CO.LTD, Republic of Korea).

## Ceramic-epoxy composite matching layer fabrication

After ceramic particles were manually mixed into the epoxy, the mixture was spread thinly on the polished piezo-ceramic or glass substrate surface. During pump-down, the applied paste was degassed in a vacuum chamber for 30 minutes under a pressure of less than 1 mTorr. The curing process proceeded in two stages, the first conducted at room temperature for 48 hours, followed by the second, conducted at 45 °C for 2 hours. The cured composite material was then lapped to the designed thickness and gradually and carefully wet-polished with a sequence of increasingly fine sandpapers.

## Transducer performance measurements

To evaluate the transducer's performance, the electrical input impedance and the two-way pulse-echo were measured. An impedance analyzer (E4990A, KEYSIGHT technology, USA) measured the electrical input impedance from 20 Hz to 60 MHz, and a pulse receiver (DPR 500, Imaginant Inc. USA) was used for pulse-echo testing. The echo response of X-cut quartz immersed in water was measured at a distance of 3 mm. The transmitted energy was 3 μJ, and the receiver gain was 0 dB.

## Thicknesses of matching layers and adhesive products

The proposed TUT uses a 7.5-MRayl $SiO_2$ micro-powder composite with a thickness of 31 μm as the 1st front matching layer and a 3.8-MRayl $SiO_2$ micro-powder composite with a thickness of 25 μm as the back-side matching layer. A clear urethane resin layer with a thickness of 18 μm and an acoustic impedance of 2.4 MRayl (Crystal Clear 202, Smooth-On, Inc., USA) forms the 2nd front matching layer. Both the proposed and conventional TUTs employ 200-nm thick indium tin oxide (ITO) electrodes on the front and back surfaces. The design of the conventional OUT is well described in[16]. The thicknesses of the 1st and 2nd front matching layers of the conventional OUT are 16 and 18 μm, respectively. The ITO electrodes are connected at four points to minimize resistance. The four corners of the front and back side matching layers are removed to connect the transducer stack to a 38 American wire gauge coaxial cable. The front side of the LNO (lithium niobate) crystal is connected to a ground wire via the brass housing ring, and the back side of the LNO is connected to the signal wire via the conductive epoxy ring (E-solder 3022, Von Roll, Switzerland). The back side of the transducer stack is filled with a clear urethane resin (Crystal Clear 202, Smooth-On, Inc., USA).

## Ultrasound and photoacoustic imaging system and experiment protocol

Figure 4a shows the overall configuration of the imaging system used in this study. A custom-designed imaging head and collimator were combined on the platform of a high-speed linear stage (NT55V65, IKO, Japan). The imaging head used a commercial collimator (FC230FC-C, Thorlabs, USA) for photoacoustic imaging (PAI). The high-speed linear stage was mounted on a manual vertical stage (BXFM-F, Olympus, Japan), and the entire assembly was attached to the platform of a low-speed linear stage (PLS-85, Physik Instrument, Germany). Custom-designed parts of the PAM body were printed with a 3D printer (MK3S, PRUSA, USA). Two protected silver front-surface mirrors (PF05-03-P01, Thorlabs, USA) were arranged in the middle of the imaging head, and a 1 cm right-angle prism mirror (MRA10-P01, Thorlabs, USA) was placed on the bottom of the imaging head to adjust the angle of the input laser beam path. A proposed TUT sample and a parabolic mirror (MPD00M9-P01, Thorlabs, USA) were mounted on the bottom of the imaging head to focus the laser beam and the acoustic beam simultaneously. For photoacoustic signal excitation, a nanosecond 1064 nm laser (SOL DPSS 1064 nm 20 W, BrightSolutions, Italy) was coupled to a multimode fiber (F-MCC-T, Newport Crop, USA) via an objective lens. The electrical signal output from the TUT was amplified by a pulser/receiver (5072PR, Olympus, Japan) and collected by a digitizer (ATS

9350, Alarzartech, Canada). System operation sequences were controlled by LabVIEW-based software (LabView 2017, National Instruments, USA) and a data acquisition board (NI-PCIe 6321, National Instruments, USA).

To transmit acoustic waves between the US transducer and the imaged object, the imaging head was placed in a custom-made water tank. The bottom of the tank had a 60 × 80 mm window covered by a thin plastic wrap. A commercial lab jack (S63080, Fisher Scientific, USA) was used to adjust the vertical distance between the water tank and the imaging subject. An acoustic matching gel (Ecosonic, SANIPIA, Republic of Korea) was applied between the plastic wrap and the imaged object. All imaging experiments were performed with lateral and elevation step sizes of 50 μm, using laser and US repetition rates of 1 kHz (B-scan rate: 1 Hz for 50 mm). A digital power meter (PM100D, Thorlabs, USA) and a photodiode power sensor (S130C, Thorlabs, USA) were used to measure optical power. The object-side irradiation energy of the 1064 nm light was 220 μJ. We used the same laser pulse energy in the chicken tissue phantom and the in-vivo experiments. The optical focal size was measured using a beam profiler (SP928, Ophir-Spiricon, USA). The estimated focal diameter was 650 μm, and the irradiation energy density at the focal point was 66 mJ/cm$^2$, which is below the American National Standards Institute (ANSI) safety limit (100 mJ/cm$^2$)[47]. In chicken breast tissue experiments, the beam spot diameter measured on the tissue surface was 1.36 mm, and the surface irradiation energy density was 15 mJ/cm$^2$. A sampling rate of 250 MS/s was used for data acquisition, and a Hilbert transform was applied to detect the envelope of the radio frequency data after applying a 1 MHz cut-off high pass filter. In the PA MAP image below 0.6 mm in Fig. 6b, a 10 MHz cut-off high-pass filter was used to remove large vessels from the image. For image visualization, all images are displayed as log-scaled colormaps. Maximum projection and cross-section images used the dynamic range indicated on figures, and average projections used a 10 dB dynamic range. All projection images and movies were created using 3D PHOVIS[48], a MATLAB-based 3D visualization software.

## Imaging phantom preparation

To measure the resolutions of ultrasound imaging (USI) and PAI, we used a 20 μm diameter tungsten wire target and a 6 μm diameter carbon fiber target, respectively. We mounted the wire and fiber between two slide glasses fixed on a flat plate. To measure the SNR of both imaging modalities, we used twenty 0.5 mm diameter pencil leads positioned horizontally at intervals of 1 mm within the chicken breast tissue. The pencil leads were precisely positioned by a 3D-printed frame.

## In-vivo imaging experiments

For mouse imaging, we followed guidelines approved by the Institutional Animal Care and Use Committee (IACUC) of Pohang University of Science and Technology (POSTECH), POSTECH-2022-0043. This research used twelve mice (POSTECH Biotech Center; BALB/c, female, 15–25 g, 6 weeks old). All mice were anesthetized with a gas system (VIP3000, Midmark, USA) containing isoflurane and oxygen gas, and the anesthesia was maintained throughout the experiment. Before conducting the imaging experiment, we depilated the mouse with a hair removal cream (Nair, Church and Dwight Co., Inc., USA). To prevent hypothermia-induced death, we maintained the mouse's body temperature by controlling a heating pad on the customized imaging stage. The mouse was waked from anesthesia after the in-vivo imaging and examined for physical injury and sequelae. We did not observe any burns caused by the laser in any of the mice, and they all acted normally afterward, such as responding to food within 1 hour. After all procedures, every mouse was sacrificed according to a method approved by the IACUC of POSTECH.

For human palm imaging, we followed regulations and guidelines approved by the Institutional Review Board (IRB) of POSTECH, PIRB-2022-E026. We recruited a healthy volunteer (male/male 1 person, 26 years, self-reported) for USI and PAI of a palm. After sufficiently explaining the experiment, we obtained informed consent from the subject. All participants in this experiment wore safety glasses and flame-retardant clothes for protection from the laser. We supported the subject's palm with a custom imaging stage to help maintain a fixed position during scanning. The subject did not experience any pain from the laser, and no visible burns were observed. There was no participant compensation.

## Preparation of IR-1048 dye solution and in-vivo experiments

The IR-1048 (CAS Number: 155613-98-2, Merck, Germany) dye was prepared by first dissolving it in dimethyl sulfoxide (DMSO), followed by dispersion using a sonicator (POWERSONIC 605, Hwashin instrument, Republic of Korea). Finally, the dye-DMSO solution was mixed with phosphate-buffered saline (PBS). The concentration of the dye was 100 μg/mL, and the volume fraction of PBS in the total solution was 98%. For mouse intravenous injection, blood vessels were dilated using a heating device and stabilized with a restrainer (RESTRAINER, JEUNG DO BIO and PLANT CO. LTD, Republic of Korea). Subsequently, 100 μL of the prepared solution was injected into the lateral tail vein, and the mouse was immediately anesthetized. The imaging protocol was subsequently carried out as described in the in-vivo imaging experiments section. After the experiment, that the mice behaved normally, engaging in activities such as grooming and eating food, until they were ultimately euthanized. A total of three mice were used to observe perfusion kinematics.

## Ex-vivo validation of IR-1048 dye accumulated in the liver

The experimental and control groups each were assigned three mice. In the experimental group, the mice were intravenously injected with IR-1048 total solution and immediately anesthetized. After 30 minutes, the anesthetized mice were euthanized, and the livers were promptly extracted. In the case of the control group, all liver procedures were performed in the same manner as in the experimental group, except for the intravenous injection. All extracted livers were meticulously cleaned of surface blood using medical gauze and subsequently transferred to a polyethylene stand. Subsequently, US gel was applied, and images were captured using the same methodology employed for in-vivo imaging.

## Fabrication of a ring-shaped ultrasound transducer

A 20 MHz ring-shaped ultrasound transducer (RUT) was fabricated in two steps: (1) fabricating the acoustic stack and (2) housing the acoustic stack. The acoustic stack, used to transmit and receive the US signals, consisted of a piezo layer (LNO, Boston Piezo-Optics, Bellingham, MA, USA), a matching layer (a 2–3 μm thickness of silver epoxy, Sigma Aldrich, St. Louis, MO, USA), and backing layer (E-Solder 3022, Von Roll, Schenectady, NY, USA). The thicknesses of the matching, piezo, and backing layers were fixed as 21, 157, and 2000 μm, respectively, and Cr and Au were deposited between each layer to electrically connect the layers. The laminated stack was machined into a ring with a 5.6 mm inner diameter and a 12.7 mm outer diameter. The ring-shaped acoustic stack was fixed on the pre-designed holder by epoxy (Epotek301, Epoxy Technology, Billerica, MA, USA) and the SMA cable's signal and ground wires were connected to the acoustic stack and holder, respectively. Finally, the fabricated transducer was coated with 26 μm Parylene-C.

## Reporting summary

Further information on research design is available in the Nature Portfolio Reporting Summary linked to this article.

## Data availability

The data that support the findings of this study are available from the corresponding author upon request.

## Code availability

Original codes for a MATLAB-based KLM model simulator and MATLAB script files for generating simulation data in this study are provided in the supplementary source data file.

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

## Acknowledgements
We thank J. Ballard for proofreading the manuscript with helpful comments, and I. Han, Y. Kim, and the Smart Soft Materials Lab in POSTECH for the help with the dynamic viscosity measurement. This work was supported by the National Research Foundation of Korea (NRF) grant funded by the Korea government(MSIT) (2023R1A2C3004880, RS-2023-00212322, 2022R1A3B1077354), Basic Science Research Program through the National Research Foundation of Korea (NRF) funded by the Ministry of Education(2020R1A6A1A03047902), National R&D Program through the National Research Foundation of Korea(NRF) funded by Ministry of Science and ICT (2021M3C1C3097624), (9991007019, KMDF_PR_20200901_0008) funded by the Ministry of Trade, and BK21 FOUR (Fostering Outstanding Universities for Research) project.

## Author contributions
The work was conceptualized by S.C., M.K. and C.K. S.C., M.K. and H.K. discussed detailed ceramic powder slurry processing methods and ultrasound transducer fabrication. S.C. and M.K. conducted all simulation and measurement processes separately and validated data by cross-checking. S.C. wrote software for acoustic and electrical simulations, and M.K. wrote software for rheological estimation and optical simulation. S.C. and M.K. fabricated TUTs, and H.K. assisted in the fabrication process. Y.K fabricated ring ultrasound transducer. S.C. designed and built the hardware for an imaging system, while J.A. developed the image acquisition software for the system. S.C., M.K., J.L., and J.P. conducted animal experiments and W.K. and C.K. assisted in the experimental protocol setting process. S.C. and M.K. conducted human palm imaging experiments and C.K. assisted in the experimental protocol setting process. S.C. and M.K. conducted phantom experiments. S.C. and M.K. processed the imaging experiment data. All processes were supervised by C.K. S.C., M.K. and C.K. drafted and revised the manuscript, and all authors discussed the results.

## Competing interests
Chulhong Kim, Seonghee Cho and Joongho Ahn have financial interests in OPTICHO, which, however, did not support this work. The remaining authors declare no other competing interests.
