## [Peer Review File · Nature Communications]

An ultrasensitive and broadband transparent ultrasound transducer for ultrasound and photoacoustic imaging in-vivoREVIEWER COMMENTS

Reviewer #1 (Remarks to the Author):

The manuscript by Cho et al. presents a groundbreaking development of a transparent ultrasound transducer, which exhibits exceptional optical and acoustic performance. The authors achieved this milestone by utilizing a newly developed SiO_2 /epoxy composite adhesive, which possesses a high acoustic impedance, as the matching and backing materials for the transducer. Consequently, the transparent transducer demonstrates comparable bandwidth and sensitivity to conventional non-transparent transducers. This advancement is of significant importance for the technological progress of transparent transducers and has the potential to revolutionize hybrid photoacoustic and ultrasound imaging, eliminating the need for bulky optical/acoustic beam combiners. Furthermore, it holds critical implications for the integration of other traditional optical imaging modalities like fluorescence imaging or optical coherence tomography.

The achievements made by the authors in this study represent a noteworthy leap forward from their previous work (reference 3)]. The manuscript is well-written, and the results presented are convincing. The technical details provided are sufficient for comprehending the fabrication process. The authors meticulously conducted proof-of-concept studies on small animal models and humans, and the translational potential of this technology for clinical applications is highly promising. Therefore, I strongly recommend the publication of this exceptional manuscript to the broad audience of Nature Communications.

While the manuscript is commendable, I have several major and minor comments that the authors should consider when improving the manuscript further.

Major comments:

1. It is crucial to compare the novel transparent ultrasound transducer (TUT) with conventional non-transparent ultrasound transducers, as the latter typically serve as the gold standard for both photoacoustic and ultrasound imaging. Although Extended Figure 1 aims to compare the acoustic impedance of various transparent transducers reported

previously, this comparison appears incomplete since transparent transducers often exhibit inferior performance in terms of bandwidth and sensitivity. I recommend conducting a similar comparison with non-transparent transducers specifically published for photoacoustic imaging, particularly those employed in acoustic-resolution photoacoustic microscopy (AR-PAM).

2. The justification for using TUT in acoustic-resolution photoacoustic microscopy (AR-PAM) is not well-established, despite its clear suitability for optical-resolution photoacoustic microscopy (OR-PAM). The working distance for AR-PAM is typically limited by the acoustic lens (or the parabolic mirror in this work). Another viable option for AR-PAM could be a ring-shaped non-transparent transducer. I suggest the authors perform additional comparisons, including theoretical modeling, to evaluate the acoustic performance of a ring-shaped transducer in comparison to the new TUT. It is possible that the central opening of the ring transducer may increase the sidelobes of the acoustic field, potentially compromising lateral resolution.

3. Although the animal and human experiments conducted in this study were well-executed, focusing on tissue morphology and blood vessel imaging, there is an opportunity to explore the full potential of this hybrid imaging system. I recommend incorporating additional demonstrations of the functional and molecular imaging capabilities of this powerful system. For instance, the manuscript could benefit from showcasing applications such as tracking drug biodistribution, mapping temperature during photothermal therapy, or quantifying tumor hypoxia. By including these additional demonstrations, the manuscript would significantly enhance its impact.

Minor comments:

1. The literature research in the introduction lacks some more recent work on transparent ultrasound transducers, such as the studies by M. Chen (Photoacoustics 28, 100417) and J. Ma (Photoacoustics 30, 100482). I suggest conducting a more thorough investigation to include these recent works in the literature review.

2. The previous transparent ultrasound transducer developed by Chen et al. employed transparent parylene as a matching layer. It would be beneficial for the authors to comment on

this design and provide a comparison between the two approaches.

3. It would be valuable to know the electrical impedance of the new transparent ultrasound transducer. Is it close to 50 ohms?

4. Please elaborate on the design advantages of using a parabolic mirror compared to attaching a concave acoustic lens, as employed in the authors' previous work. The parabolic mirror appears to be a significant limiting factor for spatial resolution and detection sensitivity.

5. What are the theoretical lateral and axial resolutions of the photoacoustic and ultrasound modes? How do these theoretical values compare to the experimental measurements?

6. What is the effective noise-equivalent detection sensitivity of the new transparent transducer, and how does it compare to the reported values of conventional non-transparent transducers?

7. The discussion section could be improved by including a comparison between the current transparent ultrasound transducers and other optical sensors for acoustic pressures, such as micro-ring resonators.

Addressing these major and minor comments will undoubtedly enhance the manuscript and further solidify its impact in the field of transparent ultrasound transducers and hybrid imaging systems.

Reviewer #2 (Remarks to the Author):

The authors reported the development of an ultrasensitive and broadband transparent

ultrasound transducer (TUT) that integrates optical and ultrasound components. The TUT addresses the limitations of existing transducers by utilizing transparent adhesive SiO₂/epoxy composite materials with optimized acoustic impedance. It achieves high optical transparency and demonstrates high-contrast and high-definition dual-modal ultrasound and photoacoustic imaging (PAI) in live animals and humans, with imaging depths exceeding 15 mm.

While the TUT in this study exhibits higher sensitivity and wider bandwidth compared to previously reported TUTs, the work lacks sufficient innovation when compared to previous studies. Although the TUT demonstrates better performance, the PA imaging setup remains similar to previous works. Our expectation for the application of TUTs is to bring significant improvements to all aspects of PAI. For instance, in current photoacoustic microscopy methods, the transducer and light need to move together. Is it possible to utilize TUTs to keep the transducer stationary and only scan the light, thus significantly enhancing imaging speed?

This work primarily focuses on process-level optimization of the transducer, lacking core theoretical innovation and failing to demonstrate significant advances of relevance to the field. As a result, it does not align with the scope of Nature Communications, and I recommend rejecting this manuscript.

Additional comments and suggestions are provided below:

1. The improvement in acoustic impedance of the transparent matching layer and backing is a significant highlight of this work. However, the authors did not mention the method used for measuring the acoustic impedance. This aspect should be addressed and supplemented in the manuscript.
2. In the PAI mode, the parabolic mirror also serves to focus the laser beam. However, the lateral resolution (acoustic resolution) is still low. It would be beneficial to discuss why this is the case.

3. The method section mentions different laser fluence levels. It would be more meaningful to provide information about the energy level used in the PAI SNR test alongside the SNR values.

Reviewer #3 (Remarks to the Author):

In this paper, the authors present a transparent ultrasound transducer (TUT) with ultrahigh sensitivity and broadband. It is applied to dual-model ultrasound and photoacoustic imaging in vivo. Both modalities reach an imaging depth of >15 mm, and their depth-to-resolution ratios exceed 500 and 370, respectively.

I support the publication of this manuscript because of its overall high quality. My detailed reasoning and suggestions are presented below:

1. The developed TUT represents the state-of-the-art in the field.

The specification of this newly developed TUT is compared with the opaque ultrasound transducer (OUT) and a conventional TUT in Figure 1b-c, Extended Data Figures 1,2, and 5. This detailed (or even exhaustive) comparison presents sufficient detail. It is informative to showcase the advantage in technology and performance of the developed TUT. It will certainly help engage a broad readership.

Despite these explanations, these data/figures, in their current forms, appear to be a simple list of information. Many insights seem to be omitted, which creates some difficulties for readers in better appreciating this work. For example, what are the criteria of design? Is the current design the optimized solution? Is there a trade-off (or balance) between efficiency and complexity?

In the description of Extended Data Figure 1, can the author elaborate on the limitations of previous works and how they are reflected in the plotted data? Lines 76-77. Also, should the content of Lines 56-62 be included in Main Text (of course in a concise form)?

In addition, is the current comparison the fairest? For the conventional OUT, I believe the comparison is reasonable as this type of transducer, such as the V212-BC-RM used by the authors in this study, is technically mature and commercially available. However, it is

unclear whether the comparison to the conventional TUT really catches the ride of the latest development in this direction. Could the author confirm that the comparison of this “conventional TUT” is the best?

The same suggestion is applied to the content related to Fig. 2. There, the design criteria need to be emphasized. What are the criteria and theories that support the design? For the text in Lines 112-118, it looks like it is introducing the experimental/simulation condition.

Should it be placed before Fig. 2b? Any details about the simulation (Lines 274-275)?

Some details about the fabrication of TUT are also missing. Again, what are the criteria to select proper (or the best) materials?

2. The impact of this transducer is manifested in dual-mode ultrasound and photoacoustic imaging

The authors provide details of the demonstration of the proposed TUT. The data are technically sound, These demonstrations serve as sufficient evidence to prove the authors' claims. Considering the attractive specifications of the proposed TUT, these results indicate its importance to the field and its wide applications in the future that will move the field forward. Considering that the work is focused on the development of the TUT, I do not expect the authors to show many novel applications. However, the data presented in the manuscript can be improved.

How did the authors obtain the resolution for PA and US imaging? It is acoustic-resolution PAM, so the resolution is determined by the transducer. However, the PA's resolution is worse than that of the US. Can the authors explain it?

If possible, the authors should also demonstrate the superior performance of the proposed TUT. In this manuscript, the authors emphasize the resolution–depth ratio. So, how would this superior performance reflect on the results? Is there anything that cannot be seen by the conventional OUT and TUT that can be seen by this TUT?

3. Minor comments

Could the authors clarify the unit of MRayl is MSK or CGS?

Lines 174-175 are not clear. In the following, the authors said that the acoustic wave was generated by TUT. So saying “The imaging head directs 1064 nm laser and US beams

through the TUT ... ” seems not accurate.

Lines 222: the authors said “In a **more demonstration**, we investigated ...”. Why this experiment is important? Probably the authors consider human experiments

Overall, this is a high-quality paper, but the presentation can be made clearer. For a high-impact paper like this, I believe it is worth writing these details in the clearest possible fashion to further enhance the overall impact of this work.

List of abbreviation

Abbreviation	Paraphrase
AR-PAM	Acoustic-resolution photoacoustic microscopy
B-mode	Brightness mode
DRR	Depth-to-resolution-ratio
FOV	Field of view
LNO	Lithium niobite
OUT	Opaque ultrasound transducer
PA	Photoacoustic
PAI	Photoacoustic imaging
PAM	Photoacoustic microscopy
Q factor	Quality factor
RUT	Ring-shaped ultrasound transducer
TUT	Transparent ultrasound transducer
US	Ultrasound
USI	Ultrasound imaging

Reviewer #1:

General comments

The manuscript by Cho et al. presents a groundbreaking development of a transparent ultrasound transducer, which exhibits exceptional optical and acoustic performance. The authors achieved this milestone by utilizing a newly developed SiO₂/epoxy composite adhesive, which possesses a high acoustic impedance, as the matching and backing materials for the transducer. Consequently, the transparent transducer demonstrates comparable bandwidth and sensitivity to conventional non-transparent transducers. This advancement is of significant importance for the technological progress of transparent transducers and has the potential to revolutionize hybrid photoacoustic and ultrasound imaging, eliminating the need for bulky optical/acoustic beam combiners. Furthermore, it holds critical implications for the integration of other traditional optical imaging modalities like fluorescence imaging or optical coherence tomography.

The achievements made by the authors in this study represent a noteworthy leap forward from their previous work (reference 3)]. The manuscript is well-written, and the results presented are convincing. The technical details provided are sufficient for comprehending the fabrication process. The authors meticulously conducted proof-of-concept studies on small animal models and humans, and the translational potential of this technology for clinical applications is highly promising. Therefore, I strongly recommend the publication of this exceptional manuscript to the broad audience of Nature Communications.

While the manuscript is commendable, I have several major and minor comments that the authors should consider when improving the manuscript further.

Reply: *Thank you for your positive and constructive feedback on our manuscript.*

Comment 1: It is crucial to compare the novel transparent ultrasound transducer (TUT) with conventional non-transparent ultrasound transducers, as the latter typically serve as the gold standard for both photoacoustic and ultrasound imaging. Although Extended Figure 1 aims to compare the acoustic impedance of various transparent transducers reported previously, this comparison appears incomplete since transparent transducers often exhibit inferior performance in terms of bandwidth and sensitivity. I recommend conducting a similar comparison with non-transparent transducers specifically published for photoacoustic imaging, particularly those employed in acoustic-resolution photoacoustic microscopy (AR-PAM).

Reply: *We have added additional figures and text for a detailed comparison between transducers in Supplementary Fig. 7 and Supplementary Table 4. Further, through simulations and experiments, we have compared the performance of the prototype novel TUT with that of a custom-made ring-shaped ultrasound transducer (RUT) in detail, and the results are summarized in Supplementary Table 5 and Supplementary Figs. 8 and 9.*

For updated information, the following sentence has been inserted in the Discussion section on Page 22:

“As detailed in Supplementary Fig. 7 and Supplementary Table 4, the performance of our prototype novel TUT, including its efficiency and noise-equivalent-pressure (NEP)³¹, is compared with those of commercial transducers commonly used for the AR-PAM, a custom-made conventional TUT, a custom-made OUT, and a custom-made ring-shaped ultrasound transducer (RUT). The custom-made US transducers exhibit higher efficiency than the commercial ones. Among the custom-made US transducers, the NEP/Af is very comparable between the OUT, RUT, and prototype novel TUT. In terms of optimal coaxial alignment between the optical and acoustical beams, the custom-made RUT and prototype novel TUT are superior. However, the sidelobes of the acoustic fields are relatively severe in the RUT, and consequently, the lateral resolution of the RUT is relatively worse than that of the prototype novel TUT. The results of simulations and experiments to compare the lateral resolutions of the custom-made RUT and prototype novel TUT are summarized in Supplementary Table 5 and

Supplementary Figs. 8 and 9.”

Supplementary Fig. 7. Measured efficiency $\eta(f)$ spectra of various ultrasound (US) transducers. TUT, transparent ultrasound transducer; OUT, opaque ultrasound transducer; and RUT, ring-shaped ultrasound transducer.

Supplementary Table 4. Performance benchmarks for various ultrasound (US) transducers. TUT, transparent ultrasound transducer; OUT, opaque ultrasound transducer; and RUT, ring-shaped ultrasound transducer.

Transducer		f (MHz)	Δf (MHz)	$\eta(f)$ (dB)	Area (mm ²)	***NEP ($\mu\text{PaHz}^{1/2}$)	NEP/ Δf ($\mu\text{PaHz}^{1/2}$)
Commercial	V308-SU (Olympus, NDT)	5	4.0	-10.5	283	23	5.7
	V212-BC-RM (Olympus, NDT)	20	37.4	-19	28	187	5
	V324-SU (Olympus, NDT)	25	7.6	-13.4	28	99	13
	V214-BC-RM (Olympus, NDT)	50	39.8	-19	28	187	4.7
	V214-BC-RM+Beam combiner	50	39.6	-20	28	210	5.3
Custom-made	Conventional TUT	30	6.4	-9	31	58	9
	Conventional OUT	30	19.3	-9	31	58	3
	RUT	20	9	-11	482	18	2
	Prototype novel TUT	30	19.3	-9	31	58	3

* Bandwidth

** Efficiency at center frequency

*** Noise-equivalent-pressure

Supplementary Table 5. Simulation parameters used to compare the point spread functions of the custom-made RUT and the prototype novel TUT.

	Custom-made RUT	Prototype novel TUT
Center frequency	20 MHz	33 MHz
Aperture	Outer diameter: 12.7 mm Inner diameter: 5.6mm	5.6 mm x 5.6 mm
Focal length		15 mm
Speed of sound		1480 m/s
Spatial pulse length		3 acoustic wavelengths

Supplementary Fig. 8. Computational comparison of the acoustic point spread functions (PSFs) of the prototype novel transparent ultrasound transducer (TUT) and the ring-shaped ultrasound transducer (RUT). **a**, Simulated PSFs of seven positions, axially spaced at 0.3-mm intervals above and below the focus. **b**, Lateral line profiles of simulated PSFs. Norm. amp, normalized amplitude; and FWHM, Full-width-half-maximum.

Supplementary Fig. 9. Photoacoustic (PA) imaging of a character logo target phantom and a leaf skeleton target using the prototype novel transparent ultrasound transducer (TUT) and the ring-shaped ultrasound transducer (RUT). Mag, magnified; and Norm. amp, normalized amplitude.

The following method is added in the Method section.

Fabrication of a ring-shaped ultrasound transducer

A 20 MHz ring-shaped ultrasound transducer (RUT) was fabricated in two steps: (1) fabricating the acoustic stack and (2) housing the acoustic stack. The acoustic stack, used to transmit and receive the US signals, consisted of a piezo layer (LNO, Boston Piezo-Optics, Bellingham, MA, USA), a matching layer (a 2–3 μm thickness of silver epoxy, Sigma Aldrich, St. Louis, MO, USA), and backing layer (E-Solder 3022, Von Roll, Schenectady, NY, USA). The thicknesses of the matching, piezo, and backing layers were fixed as 21, 157, and 2,000 μm , respectively, and Cr and Au were deposited between each layer to electrically connect the layers. The laminated stack was machined into a ring with a 5.6 mm inner diameter and a 12.7 mm outer diameter. The ring-shaped acoustic stack was fixed on the pre-designed holder by epoxy (Epotek301, Epoxy Technology, Billerica, MA, USA) and the SMA cable's signal and ground wires were connected to the acoustic stack and holder, respectively. Finally, the fabricated transducer was coated with 26 μm Parylene-C.

Comment 2: The justification for using TUT in acoustic-resolution photoacoustic microscopy (AR-PAM) is not well-established, despite its clear suitability for optical-resolution photoacoustic microscopy (OR-PAM). The working distance for AR-PAM is typically limited by the acoustic lens (or the parabolic mirror in this work). Another viable option for AR-PAM could be a ring-shaped non-transparent transducer. I suggest the authors perform additional comparisons, including theoretical modeling, to evaluate the acoustic performance of a ring-shaped transducer in comparison to the new TUT. It is possible that the central opening of the ring transducer may increase the sidelobes of the acoustic field, potentially compromising lateral resolution.

Reply: Please see the responses to Comment #1.

Comment 3: Although the animal and human experiments conducted in this study were well-executed, focusing on tissue morphology and blood vessel imaging, there is an opportunity to explore the full potential of this hybrid imaging system. I recommend incorporating additional demonstrations of the functional and molecular imaging capabilities of this powerful system. For instance, the manuscript could benefit from showcasing applications such as tracking drug biodistribution, mapping temperature during photothermal therapy, or quantifying tumor hypoxia. By including these additional demonstrations, the manuscript would significantly enhance its impact.

Reply: We have conducted additional studies to monitor the perfusion kinetics of infrared dyes in live animals, and we have added the section titled "In vivo whole-body perfusion kinematics monitoring of infrared dyes" and Fig. 9 on Page 20. The detailed experimental procedures are in the *Methods* section in Page 32 (Preparation of IR-1048 dye solution and in vivo experiments and Ex-vivo validation of IR-1048 dye accumulated in the liver). Additional information is attached below.

Fig. 9. Perfusion kinematic monitoring of IR-1048 dye in live animals. *a*, Photoacoustic (PA) maximum

amplitude projection (MAP) images of a live mouse from before injection to 9 hours after injection. **b**, 2D cross-sectional PA images cut along the line on the liver. **c**, 2D cross-sectional PA images cut along the line on the bladder. **d**, In-vivo normalized PA amplitude profile in the liver and bladder regions over time after injection ($n = 3$). **e**, Ex-vivo validation of IR-1048 accumulation in the liver at 30 minutes after injection. **f**, Ex-vivo normalized PA amplitude profile in the extracted liver ($n = 3$). Error bar denotes standard error of three experiments.

In vivo whole-body perfusion kinematics monitoring of infrared dyes

The prototype novel TUT imaging system was utilized to monitor the perfusion of an infrared dye. A time-series of PA images was acquired before and after intravenous injection of IR-1048 dye solution for 9 hours (Fig. 9). Figure 9a shows ventral-view PA MAP images of the entire mouse body pre- and post-injection. A rapid signal increase is observed in the liver region at 30 mins post-injection, and then the signal gradually decreases over time, ultimately returning to its initial brightness after 6 hours. This behavior suggests that the liver quickly takes up the injected dye, and the dye then gradually clears out. The magnified areas in the blue boxes showcase an intensified signal within the liver, characterized by a distinctive granular texture that could result from dye uptake within the hepatic lobules. Unlike the liver, the bladder displays a signal increase at 3 hr and 6 hr post-injection, returning to its initial state by 9 hr post-injection. It can be inferred that the dye is quickly taken up by the liver and then slowly cleared out via the bladder. The 30-min and 3-hr images also show numerous small bright signal dots throughout the whole body, which could indicate that the dye is excreted through the skin. Figures 9b and c are cross-sectional PA images of the liver and bladder regions over time, respectively. The PA signal peaks in the liver at 30-min post-injection, and then decreases. This behavior differs from that of the bladder signal, which peaks at 3 hr post-injection and then decreases. Figure 9d illustrates the changes in the PA signal over time. In the liver, the PA signal is dramatically enhanced at 30 min after injection (5.58 ± 0.08 , a.u.), and it still displays an enhanced signal at the 3 hr mark (3.70 ± 0.18 , a.u.). However, by 6 hr, it has returned to an amplitude similar to the control (1.03 ± 0.19 , a.u.). In the bladder, the signal gradually increases up to 3 hr post-injection (1.98 ± 0.10 , a.u.) and then has returned to its initial level by 9 hr post-injection (1.15 ± 0.13 , a.u.).

Our in vivo imaging results were validated through ex vivo studies. Figure 9e shows photographs and PA MAP images of excised liver samples with and without dye injection. The dyed liver was excised at 30 min post-injection, and the control liver was removed from a healthy mouse. Although both excised livers appear similar in the photograph, the PA signals acquired from the excised liver with dye injection are significantly stronger than those from the control liver. When we quantified the average signal difference between ex vivo samples (Fig. 9f), liver samples from dye-injected mice showed an approximately 3.7-fold enhanced signal (3.65 ± 0.54 , a.u.) over that of control livers (1.00 ± 0.30 , a.u.).

Preparation of IR-1048 dye solution and in vivo experiments

The IR-1048 (CAS Number: 155613-98-2, Merck, Germany) dye was prepared by first dissolving it in dimethyl sulfoxide (DMSO), followed by dispersion using a sonicator (POWERSONIC 605, Hwashin instrument, Republic of Korea). Finally, the dye-DMSO solution was mixed with phosphate-buffered saline (PBS). The concentration of the dye was $100 \mu\text{g/mL}$, and the volume fraction of PBS in the total solution was 98%. For mouse intravenous injection, blood vessels were dilated using a heating device and stabilized with a restrainer (RESTRAINER, JEUNG DO BIO & PLANT CO. LTD, Republic of Korea). Subsequently, $100 \mu\text{L}$ of the prepared solution was injected into the lateral tail vein, and the mouse was immediately anesthetized. The imaging protocol was subsequently carried out as described in the in-vivo imaging experiments section. After the experiment, that the mice behaved normally, engaging in activities such as grooming and eating food, until they were ultimately euthanized. A total of three mice were used to observe perfusion kinematics.

Ex-vivo validation of IR-1048 dye accumulated in the liver

The experimental and control groups each were assigned three mice. In the experimental group, the mice were intravenously injected with IR-1048 total solution and immediately anesthetized. After 30 minutes, the anesthetized mice were euthanized, and the livers were promptly extracted. In the case of the control group, all liver procedures were performed in the same manner as in the experimental group, except for the intravenous injection. All extracted livers were meticulously cleaned of surface blood using medical gauze and subsequently transferred to a polyethylene stand. Subsequently, US gel was applied, and images were captured using the same methodology employed for in-vivo imaging.

Minor comment 1: The literature research in the introduction lacks some more recent work on transparent ultrasound transducers, such as the studies by M. Chen (Photoacoustics 28, 100417) and J. Ma (Photoacoustics 30, 100482). I suggest conducting a more thorough investigation to include these recent works in the literature review.

Reply: We added the following bold text to Page 3.

“The form factor challenge can be overcome by using a transparent ultrasound transducer (TUT) or a transparent optical detector, which allows for seamless USI and OI integration^{3,12,13}. For example, Park et al. demonstrated quadruple imaging modality fusion in a single form factor, and Chen et al. implemented high-speed wide-field optical-resolution PAM (OR-PAM) using a TUT, but their TUT’s acoustic performance was inferior to that of a conventional opaque ultrasound transducer (OUT)^{3,14}. Ma et al. demonstrated high-quality multi-scale PAI through a transparent, focused optical sensor, but their optical sensor could not generate ultrasound (US) images¹⁵.”

Minor comment 2: The previous transparent ultrasound transducer developed by Chen et al. employed transparent parylene as a matching layer. It would be beneficial for the authors to comment on this design and provide a comparison between the two approaches.

Reply: This was discussed in Supplementary Note 3 in the original manuscript (Supplementary Note 4 of the revised manuscript.).

Minor comment 3: It would be valuable to know the electrical impedance of the new transparent ultrasound transducer. Is it close to 50 ohms?

Reply: Yes, the electrical input impedance is close to 50 ohms within the resonant range. We have added 50-ohm lines to Supplementary Fig. 4 to emphasize this further. We have also indicated the impedance at resonance frequency of the transducers in the impedance plot.

Supplementary Fig. 4. Comparison of the simulated electric input impedances of the novel transparent ultrasound transducer (TUT), a conventional opaque ultrasound transducer (OUT), and a conventional TUT. **a**, Measured electrical impedance. **b**, Simulated electrical impedance.

Minor comment 4: Please elaborate on the design advantages of using a parabolic mirror compared to attaching a concave acoustic lens, as employed in the authors' previous work. The parabolic mirror appears to be a significant limiting factor for spatial resolution and detection sensitivity.

Reply: *Supplementary Figure 10 presents an additional comparison between our previous work and this work, in terms of the resolution, imaging range, and sensitivity, and we have added the following bolded sentence to the Discussion on Page 22.*

“Regarding the quality of acoustic resolution imaging, although we employed a parabolic mirror, which can limit the system's performance by constraining the acoustic numerical aperture (NA_A) compared to a concave mirror, our current imaging system outperforms our previous design³ by providing a greater imaging range while preserving a comparable lateral resolution. This enhancement is demonstrated in Supplementary Fig. 10.”

Supplementary Fig. 10. *Computational pulsed acoustic field comparison for an N-SF11 concave lens-focused transparent ultrasound transducer (TUT) and the novel TUT with parabolic mirror focusing. **a**, Pulse acoustic pressure field projections for the N-SF11 concave lens-focused TUT and the parabolic mirror-focused TUT. **b**, Lateral line profile for selected positions in the pressure field projection. Norm., normalized, and NA_A , acoustic numerical aperture.*

Minor comment 5: What are the theoretical lateral and axial resolutions of the photoacoustic and ultrasound modes? How do these theoretical values compare to the experimental measurements?

Reply: *We have added the following bolded text on Page 13, and provided Supplementary Table 3*

“The measured axial and lateral resolutions show a strong correlation with theoretical values. Detailed comparisons and information for calculating theoretical values are presented in Supplementary Table 3 and Supplementary Fig. 5.”

Supplementary Table 3. Theoretical lateral and axial resolutions of ultrasound (US)/photoacoustic (PA) imaging system and measured resolution.

	Ultrasound microscopy	Acoustic resolution photoacoustic microscopy
Measured resonance frequency (f_A)		33 MHz
Measured bandwidth (Δf_A)	19.59 MHz	27.46 MHz
Theoretical axial resolution	$\frac{0.88v_A}{2\Delta f_A} \approx 33 \mu m$	$\frac{0.88v_A}{\Delta f_A} \approx 47 \mu m$
Measured axial resolution	$32.6 \pm 1.9 \mu m$	$40.4 \pm 1.6 \mu m$
Lateral resolution estimated by acoustic pressure field simulation	$102 \mu m - 145 \mu m$	$142 \mu m - 200 \mu m$
Measured lateral resolution	$110.4 \pm 21.8 \mu m$	$148.4 \pm 13.5 \mu m$

Supplementary Fig. 5. Simulated horizontal slice of point spread function (PSF) generated by the square aperture for ultrasound (US) and photoacoustic (PA) imaging and lateral and diagonal line profiles of PSF

Minor comment 6: What is the effective noise-equivalent detection sensitivity of the new transparent transducer, and how does it compare to the reported values of conventional non-transparent transducers?

Reply: Please see the responses to Comment #1.

Minor comment 7: The discussion section could be improved by including a comparison between the current transparent ultrasound transducers and other optical sensors for acoustic pressures, such as micro-ring resonators.

Reply: We have added the following bolded sentences to the discussion on Page 22.

“While optical detectors such as micro-ring resonators are very sensitive in detecting acoustic signals, and they are suitable for small form factors, they are almost impossible to generate US pressure³². Although the optical detectors are good candidates for PA imaging, they are not best suited for dual-modal PA/US imaging.”

Reviewer #2:

General comments

The authors reported the development of an ultrasensitive and broadband transparent ultrasound transducer (TUT) that integrates optical and ultrasound components. The TUT addresses the limitations of existing transducers by utilizing transparent adhesive SiO₂/epoxy composite materials with optimized acoustic impedance. It achieves high optical transparency and demonstrates high-contrast and high-definition dual-modal ultrasound and photoacoustic imaging (PAI) in live animals and humans, with imaging depths exceeding 15 mm.

While the TUT in this study exhibits higher sensitivity and wider bandwidth compared to previously reported TUTs, the work lacks sufficient innovation when compared to previous studies. Although the TUT demonstrates better performance, the PA imaging setup remains similar to previous works. Our expectation for the application of TUTs is to bring significant improvements to all aspects of PAI. For instance, in current photoacoustic microscopy methods, the transducer and light need to move together. Is it possible to utilize TUTs to keep the transducer stationary and only scan the light, thus significantly enhancing imaging speed?

This work primarily focuses on process-level optimization of the transducer, lacking core theoretical innovation and failing to demonstrate significant advances of relevance to the field. As a result, it does not align with the scope of Nature Communications, and I recommend rejecting this manuscript.

Reply: *Thanks for the important comments. First and foremost, we want to emphasize that our primary objective is to provide a dual-modal PA/US imaging system maintaining each modality's best performance. As we discussed in the original manuscript, using TUTs is the most efficient way to combine optical and acoustical beams. However, the previously reported TUTs do not provide the best performance of each modality. There are three prerequisites to fabricating a practical TUT with acoustic performance comparable to that of an OUT: 1) a transparent front matching material with an acoustic impedance of 7–9 MRayl, to maximize transmission efficiency; 2) a transparent backing material of more than 5 MRayl, to eliminate ringdowns by balancing the electrical and acoustic Q factors; and 3) a firm connection between all layers, without gaps that degrade the transducer's quality. We have considered every single step carefully to satisfy all three requirements simultaneously. Building on these innovations, we have developed a broadband (63% bandwidth) and ultrasensitive TUT, while maintaining optical transparency (> 80%). Further, we have developed an ultrasensitive dual-modal PA/US imaging system and have successfully provided whole-body structural and functional PA/US images of live animals (i.e., monitoring of pharmacokinetics upon contrast-agent injection in the updated manuscript). Note that we achieved the best depth-to-resolution of 370 and a penetration depth of beyond 15 mm in AR-PAM, which are heavily benefitted by the significantly improved sensitivity of the prototype novel TUT. The enhancement of penetration depth while maintaining resolution is the most important problem in the field, and we have solved it. Whether the optical and acoustical beams move together or not, the imaging speed fundamentally depends on the acoustic speed in the tissue, the laser repetition rate, and the scanning method. Therefore, it is not necessary to improve the sensitivity to boost the imaging speed in PAM. Recent research studies show that the imaging speed in PAM is close to the theoretical limitation^{1,2}, but this is not a focus of our work. We do agree that we failed to sufficiently present the theoretical background in the original manuscript. In the updated manuscript, we have provided the theoretical background and simulation results for designing the TUTs, measuring their performance, and comparing them with other approaches in details.*

1: Kim, Jongbeom, et al. "Super-resolution localization photoacoustic microscopy using intrinsic red blood cells as contrast absorbers." *Light: Science & Applications* 8.1 (2019): 103.

2: Zhu, Xiaoyi, et al. "Real-time whole-brain imaging of hemodynamics and oxygenation at micro-vessel resolution with ultrafast wide-field photoacoustic microscopy." *Light: Science & Applications* 11.1 (2022): 138.

Additional comments and suggestions are provided below:

Comment 1: The improvement in acoustic impedance of the transparent matching layer and backing is a significant highlight of this work. However, the authors did not mention the method used for measuring the acoustic impedance. This aspect should be addressed and supplemented in the manuscript.

Reply: *To provide this information, we have included the following sentence in the "Acoustical simulation of the ceramic-epoxy composites and their experimental validation" section of the Methods on Page 26.*

"We calculated the acoustic impedance of the material by measuring the travel time of acoustic waves and obtaining the weight and volume of a half-inch diameter chip with a thickness of 2 mm. The dimensions of the sample chips were controlled to $\pm 1 \mu\text{m}$. The density of each sample was calculated using the measured volume and weight values, and we multiplied this value by the longitudinal velocity of each sample."

Comment 2: In the PAI mode, the parabolic mirror also serves to focus the laser beam. However, the lateral resolution (acoustic resolution) is still low. It would be beneficial to discuss why this is the case.

Reply: *In this paper, our objective is to demonstrate the combination of deep imaging depth and high axial resolution in the optical diffusion regime (i.e., beyond 1 mm in tissues) by utilizing a highly sensitive ultrasonic transducer and optimal illumination, and we achieved an enhanced depth-to-resolution ratio which was never reported before. Improving the optical resolution is not a focus of our work. To clarify our intention, in the "Combined ultrasound/photoacoustic microscopy system using the TUT, and its imaging results in live animals and humans" section, we have included the following bolded sentence on the Page 12:*

"Our system is specifically designed for deep tissue imaging using acoustic-resolution photoacoustic microscopy."

Comment 3: The method section mentions different laser fluence levels. It would be more meaningful to provide information about the energy level used in the PAI SNR test alongside the SNR values.

Reply: *We maintained a consistent pulse energy level of '220 μJ ' for the chicken tissue phantom SNR measurement, and in vivo experiments. To make the information clearer, we have included the following sentence on Page 30:*

"We used the same laser pulse energy in the chicken tissue phantom and the in vivo experiments"

We apologize for our error in mistakenly referring to "energy" and "energy density as " power.". To avoid confusion, we have corrected these mistakes, and now they are worded as "energy." or "energy density."

Reviewer 3:

General comments

In this paper, the authors present a transparent ultrasound transducer (TUT) with ultrahigh sensitivity and broadband. It is applied to dual-model ultrasound and photoacoustic imaging in vivo. Both modalities reach an imaging depth of >15 mm, and their depth-to-resolution ratios exceed 500 and 370, respectively.

I support the publication of this manuscript because of its overall high quality. My detailed reasoning and suggestions are presented below.

Reply: *Thank you for your positive feedback.*

Comment 1: The developed TUT represents the state-of-the-art in the field.

The specification of this newly developed TUT is compared with the opaque ultrasound transducer (OUT) and a conventional TUT in Figure 1b-c, Extended Data Figures 1,2, and 5. This detailed (or even exhaustive) comparison presents sufficient detail. It is informative to showcase the advantage in technology and performance of the developed TUT. It will certainly help engage a broad readership.

Despite these explanations, these data/figures, in their current forms, appear to be a simple list of information.

Comment 1-1: Many insights seem to be omitted, which creates some difficulties for readers in better appreciating this work. For example, what are the criteria of design? Is the current design the optimized solution? Is there a trade-off (or balance) between efficiency and complexity?

Reply: *The design requirements for the prototype novel TUT, are described in Lines 35–39 of the original manuscript. We set three design criteria to achieve these requirements, summarized as follows:*

Criterion	Requirements
Acoustic impedance	1. front matching material with an acoustic impedance of 7–9 MRayl 2. backing material of more than 5 MRayl,
Viscosity	Below 100 McPs
Optical transparency	Transparent

To offer a comprehensive understanding of the transducer's design requirements, criteria and matching material, we have included the following bolded sentence on Page 3 with additional Supplementary Figs 1 and 2, and Supplementary Notes 2 and 3.

***“The 1st prerequisite, for the front matching layer, ensures a consistent and high transmit pressure for water especially when utilized with a pure polymer with an acoustic impedance of 2–3 MRayl in a double matching structure. Here, we aim to achieve flat and high gains by utilizing double matching layers consisting of a 7.5 MRayl 1st layer with and a 2.36 MRayl 2nd layer. Detailed information on the front matching is in Supplementary Fig. 1 and Supplementary Note 2. The backing material's acoustic impedance requirement balances the acoustic and electrical Q factors. Given the frontside specifications, backing material with an acoustic impedance of 7.2 MRayl will achieve the optimal bandwidth. In US transducers with similar designs, slightly lighter damping of 5–6 MRayl, is often employed, which does not significantly unbalance the Q factors and enhances sensitivity. This study also aims for lighter damping and accordingly adopts a double-layered backing structure with an acoustic impedance of 6.1 MRayl at the resonance center. The design employs a matching layer of 3.8 MRayl to increase the effective acoustic impedance of the backside without losing transparency. Additional explanations for the backing can be found in Supplementary Note 3. To ensure a solid connection between all layers, the adhesion gap should be minimal compared to the acoustic wavelength. The most effective approach to eliminating the adhesion gap is to use the matching and backing material as an adhesive and directly cure it on the adhesion surface. Hence, this study aims to create a material with a viscosity suitable for adhesive-like bonding. A viscosity of 100 McPs is considered the upper limit because higher-viscosity materials are challenging to pour or spread evenly*”**

on a surface.

In this work, to satisfy all three requirements simultaneously, we have used experiments and simulations to devise a recipe that provides the needed acoustic, rheological, and optical properties. Supplementary Figure 2 illustrates the detailed simulation process. The new adhesive SiO₂/epoxy composite materials provide optimized optical transparency, acoustic impedance, and flowability.”

Because the contents of Supplementary Note 2 of the original manuscript are included in the revised manuscript and updated Supplementary Notes 1 and 2, Supplementary Note 2 of the original manuscript is deleted.

Additionally, updated Supplementary Notes and Figures are available below.

Supplementary Fig. 1. Gain response and pressure transmittance on the frontside of a piezo crystal

Supplementary Note 1. Effect of front layer combination on transfer pressure gain and transducer performance

Optimizing pressure amplification gain and pressure transmittance is crucial to enhancing the performance of contemporary TUTs. These factors depend on the difference in acoustic impedance between the gain medium and the transmission medium. A larger mismatch leads to higher pressure amplification but lower transfer efficiency, while a closer match results in lower pressure amplification but higher transfer efficiency. The critical challenge in TUT production is finding the right balance to

achieve a transmit pressure gain that is both flat and high. These factors can be calculated by examining the reflectance of the front half of the acoustic transmission line from the center of the piezo crystal. The pressure transmittance to the loading medium is determined by the following equation:

$$T_F = \sqrt{\frac{Z_F}{Z_C} (1 - |\Gamma_F|^2)}, \quad \Gamma_F = \frac{Z_{F_{in}} - Z_C}{Z_{F_{in}} + Z_C},$$

where T_F is pressure transmittance from the piezo crystal to the front load medium, Z_F is the acoustic impedance of the front load medium, $Z_{F_{in}}$ is the acoustic input impedance of the front half of the transducer, Z_C is the acoustic impedance of the piezo crystal, and Γ_F is the pressure reflectance caused by the front part of the transducer. Assuming that the front half of the transducer is connected to a perfectly reflective boundary, the resulting front pressure amplification gain (G_F) and transmitted pressure gain (G_{FT}) can each be expressed as the infinite sum of an infinite geometric series:

$$G_F = \frac{1}{1 - \Gamma_F}, \quad G_{FT} = G_F T_F.$$

The calculated results for several representative cases are depicted in Supplementary Fig. 1. As depicted, we have presented the normalized frequency response of gain and transmittance concerning the acoustic impedance variation of the 1st matching layer. In the left column, we show pressure amplification gain, pressure transmittance, and normalized transmitted gain. We conducted normalization based on the maximum and minimum value of the transmitted pressure gain for each acoustic impedance. Among the three illustrated cases, the 2.36 MRayl 2nd layer in the double matching structure exhibits the most consistent response in normalized transmitted gain. In this instance, when the 1st layer has an impedance below 7 MRayl, the transmittance is low, and when the 2nd layer exceeds 9 MRayl, the amplification gain significantly diminishes. These outcomes appear due to narrow bandwidth and dual-frequency characteristics, respectively. Therefore, it is reasonable for the 1st layer to fall within the range of 7–9 MRayl, and we opted for 7.5 MRayl, which shows a highly flat response.

Supplementary Note 2. Recommended acoustic impedance of the backing layer for optimum bandwidth

According to Desilet *et al.*'s¹ analysis using the Krimholtz, Leedom, and Matthaei (KLM) model, the transducer has an optimum bandwidth when the acoustic Q factor (Q_a) and electric Q factor (Q_e) are balanced, and each Q factor can be estimated as follows.

$$Q_a = \frac{\pi Z_C}{2(Z_B + Z_F)}, \quad Q_e = \frac{\pi(Z_B + Z_F)}{4k_t^2 Z_C}.$$

Here, k_t is the electromechanical coupling coefficient, and Z_B , Z_F , and Z_C represent the effective acoustic impedances of the backload, frontload, and gain medium. When water (1.5 MRayl) is the final load, the front load of our prototype novel TUT is 16.4 MRayl. For LNO, k_t is 0.49 and Z_C is 34.1 MRayl. In this case, the backload that equalizes Q_a and Q_e is 7.2 MRayl.

Supplementary Fig. 2. Scheme for finding an appropriate matching material design

Comment 1-2: In the description of Extended Data Figure 1, can the author elaborate on the limitations of previous works and how they are reflected in the plotted data? Lines 76-77. Also, should the content of Lines 56-62 be included in Main Text (of course in a concise form)?

Reply: For Lines 76-77, as we replied to Comment 1-1, in terms of magnitude, if the acoustic impedance is below the ideal zone, the transducer exhibits poor transmittance to acoustic waves, leading to reduced sensitivity and resolution. Conversely, if the acoustic impedance is above the ideal zone, the transducer experiences inadequate acoustic gain, resulting in insufficient sensitivity and resolution. Detailed information about previous works is already discussed in Supplementary Note 3 in the original manuscript. Further, to better depict the typical limitations and general characteristics of each area, we have updated Supplementary Fig. 3 (formerly Extended Data Figure 1) as shown below.

Supplementary Fig. 3. Comparisons of the effective front loads and phase angles in previously reported studies.

For Lines 56-62, we have included brief information about the typical configuration of a conventional OUT design on Page 6 as follows:

“In the conventional OUT, metal-epoxy composites are commonly used as both the 1st front matching layer and the backing layer to achieve the desired acoustic impedance and electrical conductivity, thus fulfilling acoustic and viscosity design criteria. Additional detailed information regarding a standard requirement for a conventional OUT is provided in Supplementary Note 3.”

Comment 1-3: In addition, is the current comparison the fairest? For the conventional OUT, I believe the comparison is reasonable as this type of transducer, such as the V212-BC-RM used by the authors in this study, is technically mature and commercially available. However, it is unclear whether the comparison to the conventional TUT really catches the ride of the latest development in this direction. Could the author confirm that the comparison of this “conventional TUT” is the best?

Reply: Please refer to the reply to Comment #1 of Reviewer #1. We used custom-made conventional OUTs and TUTs to compare them with our proposed novel TUT, using the most up-to-date technology available. While commercially available single-element transducers such as the V212-BC-RM and similar products demonstrate robust quality and high yield in production, they are not specifically designed or manufactured for high-quality ultrasound imaging. Unlike modern imaging ultrasound transducers that achieve broadband properties through improved matching technology, they achieve their broad bandwidth by utilizing high damping, as stated in their transducer catalog and official information. Therefore, they lack sensitivity in comparison to double-matching-based transducers. We have already conducted a comparative study between custom-made double-matching-based transducers and commercial transducers in photoacoustic research, as outlined in the following publication:

“Kim, H., Kim, J. Y., Cho, S., Ahn, J., Kim, Y., Kim, H., & Kim, C. (2022). Performance comparison of high-speed photoacoustic microscopy: opto-ultrasound combiner versus ring-shaped ultrasound transducer. *Biomedical Engineering Letters*, 12(2), 147-153.”

Comment 1-4: The same suggestion is applied to the content related to Fig. 2. There, the design criteria need to be emphasized. What are the criteria and theories that support the design? For the text in Lines 112-118, it looks like it is introducing the experimental/simulation condition. Should it be placed before Fig. 2b? Any details about the simulation (Lines 274-275)?

Reply: Please refer to our response to Comment #1-1, where we provided detailed information about the design criteria, including the grounded theory. We also addressed the experimental and simulation conditions in lines 112-118 in that response.

For lines 274-275, we utilized our custom MATLAB-based KLM model simulator for transducer simulation, which enabled us to obtain the reported values. The pulse-echo waveforms and response spectra were generated for a perfect reflection target, and the simulations did not consider signal loss in the propagation medium. We have added the following bolded text to the **Method** section on Page 25:

“A custom MATLAB-based KLM model simulator using the ABCD matrix computation scheme was built and used. The pulse-echo waveforms and response spectra were generated for a perfect reflection target, and simulation simulations did not consider signal loss in the propagation medium.”

Comment 1-5: Some details about the fabrication of TUT are also missing. Again, what are the criteria to select proper (or the best) materials?

Reply: Please see the responses to Comment #1-1.

Comment 2: The impact of this transducer is manifested in dual-mode ultrasound and photoacoustic imaging

The authors provide details of the demonstration of the proposed TUT. The data are technically sound, These demonstrations serve as sufficient evidence to prove the authors' claims. Considering the attractive specifications of the proposed TUT, these results indicate its importance to the field and its wide applications in the future that will move the field forward. Considering that the work is focused on the development of the TUT, I do not expect the authors to show many novel applications. However, the data presented in the manuscript can be improved.

Reply: *We appreciate the positive comments. We have conducted additional studies to monitor the perfusion kinetics of infrared dyes in live animals. See the response to Comment #3 of Reviewer #1.*

Comment 2-1: How did the authors obtain the resolution for PA and US imaging? It is acoustic-resolution PAM, so the resolution is determined by the transducer. However, the PA's resolution is worse than that of the US. Can the authors explain it?

Reply: *See the responses of Minor Comment #5 of Reviewer #1. Our measured resolution demonstrates excellent agreement with the theoretical values.*

Comment 2-2: If possible, the authors should also demonstrate the superior performance of the proposed TUT. In this manuscript, the authors emphasize the resolution–depth ratio. So, how would this superior performance reflect on the results? Is there anything that cannot be seen by the conventional OUT and TUT that can be seen by this TUT?

Reply: *We added the following bolded text to the manuscript and accompanying paragraphs to the manuscript on Page 23.*

“Compared to previous AR-PAM studies, the high-DRR system in this study offers enhanced performance, providing deeper penetration and larger in vivo FOV images. The prototype novel TUT employed here exhibited higher detection efficiency and lower NEP than the commercial transducers used in previously reported systems (Supplementary Fig. 3 and Supplementary Table 4). The novel TUT also works as a transparent optical window. These innovations combine to offer significantly expanded imaging capabilities. While most previous studies primarily visualized blood vessels within superficial layers in mice, such as the peritoneum and skin, reaching no deeper than 3 mm below the skin surface³³⁻³⁷, our system can visualize even common organ structures in mice. In a previous study, an imaging system utilizing dark-field illumination and a 5 MHz frequency transducer could penetrate deeply enough to visualize common organ structures in mice, but it sacrificed resolution to do so³⁸. Moreover, its penetration depth was inferior to that of our system (10.3 mm vs. 15 mm). Still another dark-field illumination approach, using a 30 MHz transducer and a 1064 nm wavelength, demonstrated shallower penetration into a chicken breast phantom (11 mm vs. 13 mm) and comparable spatial resolution to our system (57 μm vs. 40 μm axially, 130 μm vs. 148 μm laterally), using about 8 times higher laser energy (1.8 mJ vs. 220 μJ)³⁹. Its phantom experiment using a black-tape target demonstrated relatively deep penetration compared to other studies. However, it provided images of in vivo black ink injections only at limited depth, which presented challenges in visualizing signals with an insufficient SNR from deeper mouse tissue components. In contrast, we visualized not only the PA signal from blood vessels but also the normal tissue of mice, as demonstrated in the B-scan images presented in Figs.6 and 9. In the case of human palm imaging, while other studies primarily visualized shallow blood vessels within 2 mm depth^{33,40}, our study successfully distinguished and visualized the skin, subcutaneous vessels, veins, and arteries.”

Minor comment 1: Could the authors clarify the unit of MRayl is MSK or CGS?

Reply: *It is MKS. We added the following bolded text to the manuscript.*

“1 Rayl= 1 Pa·s·m⁻¹= 1 kg/(m²·s)”

Minor comment 2: Lines 174-175 are not clear. In the following, the authors said that the acoustic wave was generated by TUT. So saying “The imaging head directs 1064 nm laser and US beams through the TUT ... ” seems not accurate. .

Reply: *We revised the sentence to the manuscript on Page 13:*

“The imaging head consists of two angle-adjustable mirrors and a fixed right-angle mirror prism, which direct the output of a 1064 nm laser through the TUT. A parabolic mirror simultaneously focuses both the laser beam, after it passes through the TUT, and the US beam.”

Minor comment 3: Lines 222: the authors said “In a more demonstration, we investigated ...”. Why this experiment is important? Probably the authors consider human experiments

Reply: *We have added the following sentences in Page 18.*

“Imaging blood vessels in human extremities are important in various diseases such as diabetes and Raynaud’s diseases.”

REVIEWERS' COMMENTS:

Reviewer #1 (Remarks to the Author):

I thank the authors for the comprehensive revision of their manuscript, which has now addressed all of my comments/questions. I also believe that the other reviewers' comments are adequately addressed. I fully support the publication of this excellent work.

Reviewer #2 (Remarks to the Author):

From an innovation perspective, the authors have not yet convinced me. In my opinion, their response still appears to be more of a process improvement rather than a theoretical innovation or a technological revolution.

The author mentioned an ultrasensitive dual-modal PA/US imaging system as part of the innovation, but this system seems to be similar to the one described in their previous publication (<https://doi.org/10.1073/pnas.1920879118>). Please elaborate on the differences between the systems in these two works.

I still hold my opinion that this article is not suitable for publication in Nature Communications.

Reviewer #3 (Remarks to the Author):

The authors have well addressed my comments. I recommend publishing this manuscript.

Reviewer #1 (Remarks to the Author):

I thank the authors for the comprehensive revision of their manuscript, which has now addressed all of my comments/questions. I also believe that the other reviewers' comments are adequately addressed. I fully support the publication of this excellent work.

***Reply:** We sincerely appreciate your feedback. Thanks to your valuable comments, the foundation of the manuscript has been strengthened. We are delighted to address all your suggestions.*

Reviewer #2 (Remarks to the Author):

From an innovation perspective, the authors have not yet convinced me. In my opinion, their response still appears to be more of a process improvement rather than a theoretical innovation or a technological revolution.

The author mentioned an ultrasensitive dual-modal PA/US imaging system as part of the innovation, but this system seems to be similar to the one described in their previous publication (<https://doi.org/10.1073/pnas.1920879118>). Please elaborate on the differences between the systems in these two works.

I still hold my opinion that this article is not suitable for publication in Nature Communications.

***Reply:** The most significant difference between the current and previous works is the significant improvement on the performance of transparent ultrasound transducer (TUT), compatible to the existing opaque ultrasound transducers. Consequently, this enables high-contrast and high-definition dual-modal ultrasound and photoacoustic imaging in live animals and humans. Both modalities reach an imaging depth of > 15 mm, with depth-to-resolution ratios (DRRs) exceeding 500 and 370, respectively. Particularly, this DRR in acoustic-resolution photoacoustic microscopy has been never achieved in any other previous works. In our previous work, the acoustic impedance mismatch between the layers composing the TUT resulted in unfavorable resonance properties, leading to limited resolution and sensitivity. In the current study, we addressed the acoustic mismatch issue in the TUTs by introducing novel layer materials for front and backside matching. This approach overcame the resonance problems and consequently improved the resolution and sensitivity, enhancing the overall performance of the TUT.*

In the introduction, we highlighted the issue of acoustic mismatch, showcasing our commitment to addressing this challenge. In the results section, we demonstrated high sensitivity and wide bandwidth, and illustrated our approach's potential value by achieving a high DRR in in-vivo ultrasound and photoacoustic imaging. Finally, in the discussion section, we emphasized the importance of our achievement of sensitivity, resolution, and DRR.

For minor technical distinctions in the imaging systems, we have already delineated the variance between the concave lens utilized in our prior research and the parabolic mirror employed in this study. This information can be found in lines 295–299 in Page 12 and Supplementary Figure 10.

Reviewer #3 (Remarks to the Author):

The authors have well addressed my comments. I recommend publishing this manuscript.

***Reply:** Thank you very much for bringing attention to the story we overlooked in our first submission. Your input has significantly improved the manuscript, and we sincerely appreciate your valuable contribution.*